# Within-host evolutionary dynamics of seasonal and pandemic human influenza A viruses in young children

Alvin X Han[1†]*, Zandra C Felix Garza[1†], Matthijs RA Welkers[1†], René M Vigeveno[1], Nhu Duong Tran[2], Thi Quynh Mai Le[2], Thai Pham Quang[2], Dinh Thoang Dang[3], Thi Ngoc Anh Tran[4], Manh Tuan Ha[4], Thanh Hung Nguyen[5], Quoc Thinh Le[5], Thanh Hai Le[6], Thi Bich Ngoc Hoang[6], Kulkanya Chokephaibulkit[7], Pilaipan Puthavathana[7], Van Vinh Chau Nguyen[8], My Ngoc Nghiem[8], Van Kinh Nguyen[9], Tuyet Trinh Dao[9], Tinh Hien Tran[7,10], Heiman FL Wertheim[10,11,12], Peter W Horby[12,13], Annette Fox[13,14,15], H Rogier van Doorn[12,13], Dirk Eggink[1,16‡], Menno D de Jong[1‡], Colin A Russell[1‡]*

[1]Department of Medical Microbiology & Infection Prevention, Amsterdam University Medical Center, Amsterdam, Netherlands; [2]National Institute of Hygiene and Epidemiology, Hanoi, Viet Nam; [3]Ha Nam Centre for Disease Control, Ha Nam, Viet Nam; [4]Children's Hospital 2, Ho Chi Minh city, Viet Nam; [5]Children's Hospital 1, Ho Chi Minh city, Viet Nam; [6]Vietnam National Children's Hospital, Hanoi, Viet Nam; [7]Siriraj Hospital, Mahidol University, Bangkok, Thailand; [8]Hospital for Tropical Diseases, Ho Chi Minh city, Viet Nam; [9]National Hospital for Tropical Diseases, Hanoi, Viet Nam; [10]Oxford University Clinical Research Unit, Ho Chi Minh city, Viet Nam; [11]Radboud Medical Centre, Radboud University, Nijmegen, Netherlands; [12]Nuffield Department of Medicine, University of Oxford, Oxford, United Kingdom; [13]Oxford University Clinical Research Unit, Hanoi, Viet Nam; [14]Peter Doherty Institute for Infection and Immunity, University of Melbourne, Melbourne, Australia; [15]WHO Collaborating Centre for Reference and Research on Influenza, Melbourne, Australia; [16]Centre for Infectious Disease Control, National Institute for Public Health and the Environment, Bilthoven, Netherlands

*For correspondence:
x.han@amsterdamumc.nl (AXH);
c.a.russell@amsterdamumc.nl
(CAR)

[†]These authors contributed
equally to this work
[‡]These authors also contributed
equally to this work

Competing interests: The
authors declare that no
competing interests exist.

Reviewing editor: Sarah E
Cobey, University of Chicago,
United States

**Abstract** The evolution of influenza viruses is fundamentally shaped by within-host processes. However, the within-host evolutionary dynamics of influenza viruses remain incompletely understood, in part because most studies have focused on infections in healthy adults based on single timepoint data. Here, we analyzed the within-host evolution of 82 longitudinally sampled individuals, mostly young children, infected with A/H1N1pdm09 or A/H3N2 viruses between 2007 and 2009. For A/H1N1pdm09 infections during the 2009 pandemic, nonsynonymous minority variants were more prevalent than synonymous ones. For A/H3N2 viruses in young children, early infection was dominated by purifying selection. As these infections progressed, nonsynonymous variants typically increased in frequency even when within-host virus titers decreased. Unlike the short-lived infections of adults where de novo within-host variants are rare, longer infections in young children allow for the maintenance of virus diversity via mutation-selection balance creating potentially important opportunities for within-host virus evolution.

## Introduction

Influenza A viruses (IAV) are some of the most prevalent human respiratory pathogens, infecting hundreds of millions of people worldwide each year. Because of the high error rates of the viral RNA polymerase complex, de novo mutants are generated as the viruses replicate within infected hosts (*Andino and Domingo, 2015*). However, the emergence of these variants within host does not mean that they will become the majority variant within the infected host or be transmitted between hosts. The evolution of IAVs is the product of a complex mosaic of evolutionary processes that include genetic drift, positive selection (*Smith et al., 2004*), transmission bottleneck effects (*Varble et al., 2014*; *McCrone et al., 2018*), and global migration patterns (*Russell et al., 2008*; *Rambaut et al., 2008*). Importantly, the resulting evolutionary dynamics can differ at the individual and population levels (*Nelson and Holmes, 2007*).

For seasonal IAVs at the global population level, antibody-mediated immune selection pressure from natural infection or vaccination positively selects for novel antigenic variants that facilitate immune escape, resulting in antigenic drift (*Smith et al., 2004*). However, at the within-host level, the role of positive selection exerted by immunity is less obvious. Several next-generation sequencing studies of typical, short-lived seasonal IAV infections in adult humans showed that intra-host genetic diversity of influenza viruses is low and dominated by purifying selection (*McCrone et al., 2018*; *Dinis et al., 2016*; *Debbink et al., 2017*; *Valesano et al., 2020*; *Sobel Leonard et al., 2016*). Additionally, large-scale comparative analyses of IAV hemagglutinin (HA) consensus sequences found limited evidence of positive selection on HA at the individual level regardless of the person's expected influenza virus infection history (*Han et al., 2019*). Importantly, these studies focused on virus samples from only one or two timepoints, mostly early in infection, limiting the opportunities to study how virus populations evolved over the course of infection.

Separate from seasonal IAVs, zoonotic IAVs constantly pose new pandemic threats. Prior to becoming human-adapted seasonal strains, IAVs are introduced into the human population from an animal reservoir through the acquisition of host-adaptive mutations, sometimes via reassortment, resulting in global pandemics such as the 2009 swine influenza pandemic (*Smith et al., 2009*). In the 2009 pandemic, global virus genetic diversity increased rapidly during the early phases of the pandemic as a result of rapid transmissions in the predominantly naïve human population (*Su et al., 2015*). Over subsequent waves of the pandemic, host-adapting mutations that incrementally improved viral fitness and transmissibility in humans of A/H1N1pdm09 viruses emerged (*Elderfield et al., 2014*), eventually reaching fixation in the global virus population (*Nogales et al., 2018*).

At the individual level, the within-host evolutionary dynamics of the pandemic A/H1N1pdm09 virus, particularly in the early stages of the 2009 pandemic, have been relatively underexplored. To date, the only within-host genetic diversity analysis of A/H1N1pdm09 viruses during the initial phase of the pandemic was based on mostly single-timepoint samples collected within ~7 days post-symptom onset (*Poon et al., 2016*). Despite initial findings of high within-host diversity and loose transmission bottlenecks (*Poon et al., 2016*), these results were later disputed due to technical anomalies and subsequent reanalyses of a smaller subset of the original data found that intra-host genetic diversity of the pandemic virus was low and comparable to levels observed in seasonal IAVs (*Xue and Bloom, 2019*; *Poon et al., 2019*). It remains unclear how frequently host-adaptive mutations appear within hosts infected by a pandemic IAV and if these mutants are readily transmitted between individuals.

Here, we deep-sequenced 275 longitudinal clinical specimens sampled from 82 individuals residing in Southeast Asia between 2007 and 2009 that were either infected with seasonal A/H3N2 or pandemic A/H1N1pdm09 viruses. By analyzing minority variants found across the whole IAV genome, we characterized the evolutionary dynamics of within-host virus populations in these samples collected up to 2 weeks post-symptom onset.

## Results

### Study participants

The A/H3N2 virus samples were collected from 51 unlinked individuals as part of an oseltamivir dosage trial (*South East Asia Infectious Disease Clinical Research Network, 2013*; *Koel et al., 2020*).

48 of the 51 A/H3N2 virus-infected individuals were young children (median age = 2 years; inter-quartile range [IQR] = 2–3 years) at the time of sampling and most had low or no detectable anti-influenza virus antibody titers on days 0 and 10 post-symptom onset (*Koel et al., 2020*). Given that young children are substantial contributors to influenza virus transmission (*Worby et al., 2015*; *Viboud et al., 2004*), the samples analyzed here offer a valuable opportunity to investigate the within-host IAV evolutionary dynamics in this key population. The A/H1N1pdm09 virus specimens were collected from 32 individuals up to 12 days post-symptom onset. These individuals include both children and adults (median age = 10 years; IQR = 4–20 years) infected during the first wave of the pandemic in Vietnam (July–December 2009). 15 of the 32 individuals (including six index patients) were sampled in a household-based influenza cohort study (*Horby et al., 2012*). The remaining 16 unlinked individuals were hospitalized patients that were involved in two different osel-tamivir treatment studies (*South East Asia Infectious Disease Clinical Research Network, 2013*; *Hien et al., 2010*) Details of all study participants are described in the respective cited studies and *Supplementary file 4*.

## Genetic diversity of within-host virus populations

We used the number of minority intra-host single-nucleotide variants (iSNVs; $\geq 2\%$ in frequencies) to measure the levels of genetic diversity of within-host IAV populations. Similar to previous studies (*McCrone et al., 2018*; *Dinis et al., 2016*; *Debbink et al., 2017*; *Sobel Leonard et al., 2016*), within-host genetic diversity of human A/H3N2 virus populations was low (median = 11 iSNVs, IQR = 7–16; *Figure 1A*). Within-host genetic diversity of pandemic A/H1N1pdm09 virus populations was also low, with a median number of 21 iSNVs (IQR = 13.5–30.0; *Figure 1B*) identified. Cycle threshold (Ct) values, and thus likely virus shedding, correlated with the number of days post-symptom onset for both IAV subtypes (A/H3N2: Spearman's $\rho = 0.468$, $p = 1.38 \times 10^{-10}$; A/H1N1pdm09: $\rho = 0.341$, $p = 0.048$; *Figure 1C, D*). The number of iSNVs observed in within-host A/H3N2 virus populations weakly correlated with days since onset of symptoms in patients ($\rho = 0.463, p = 2.22 \times 10^{-10}$) and Ct values ($\rho = 0.508, p = 1.20 \times 10^{-12}$), suggesting that as infection progresses genetic variants accumu-late within-host even as virus population size decreases (*Figure 1A*). On the other hand, there was no significant correlation between the number of iSNVs observed in within-host A/H1N1pdm09 virus populations and Ct values ($\rho = 0.198, p = 0.21$) or days post-symptom onset ($\rho = -0.021, p = 0.91$; *Figure 1B*).

## Within-host evolutionary rates of IAV

To investigate within-host evolutionary dynamics, empirical rates of synonymous, nonsynonymous, and premature stop codon (i.e., nonsense) iSNVs were calculated by normalizing the summation of observed iSNV frequencies with the number of available sites and time since symptom onset (see Materials and methods). The overall within-host evolutionary rates of A/H3N2 viruses observed here are in the same order of magnitude ($< \sim 10^{-5}$ divergence per site per day) as those reported in previ-ous within-host seasonal influenza virus evolution studies (*Figure 2A*; *Xue and Bloom, 2020*). Synon-ymous evolutionary rates were significantly higher than nonsynonymous rates during the initial phase of A/H3N2 virus infections (*Figure 2A*), primarily in the polymerase complex and HA genes (*Figure 2A*, *Figure 2—figure supplement 1*, and *Figure 3—figure supplement 1*). Importantly, nonsynonymous variants gradually accumulated, increasing in rates around 4 days post-symptom onset to similar levels relative to synonymous rates. To ensure that this temporal trend was not due to aggregated effects across multiple individuals, we performed linear regression on the computed evolutionary rates for each A/H3N2-infected individual with at least three sampling timepoints (n = 39). Nonsynonymous evolutionary rates were positively correlated against time for 25/39 indi-viduals (64%; *Figure 2—figure supplement 3*). In contrast, synonymous evolutionary rates were neg-atively correlated against time for 27 (69%) individuals.

Consolidating over all samples, most nonsynonymous variants were found in the nucleoprotein (NP) and neuraminidase (NA) gene segments (nonsynonymous to synonymous variant [NS/S] ratios = 1.69 [NP] and 1.32 [NA], whereas NS/S ratios were $\leq 1$ for all other gene segments; *Figure 2—figure supplement 1* and *Supplementary file 1*). While nonsynonymous NA mutations associated with oseltamivir resistance were positively selected for a subset of individuals in response to the antiviral

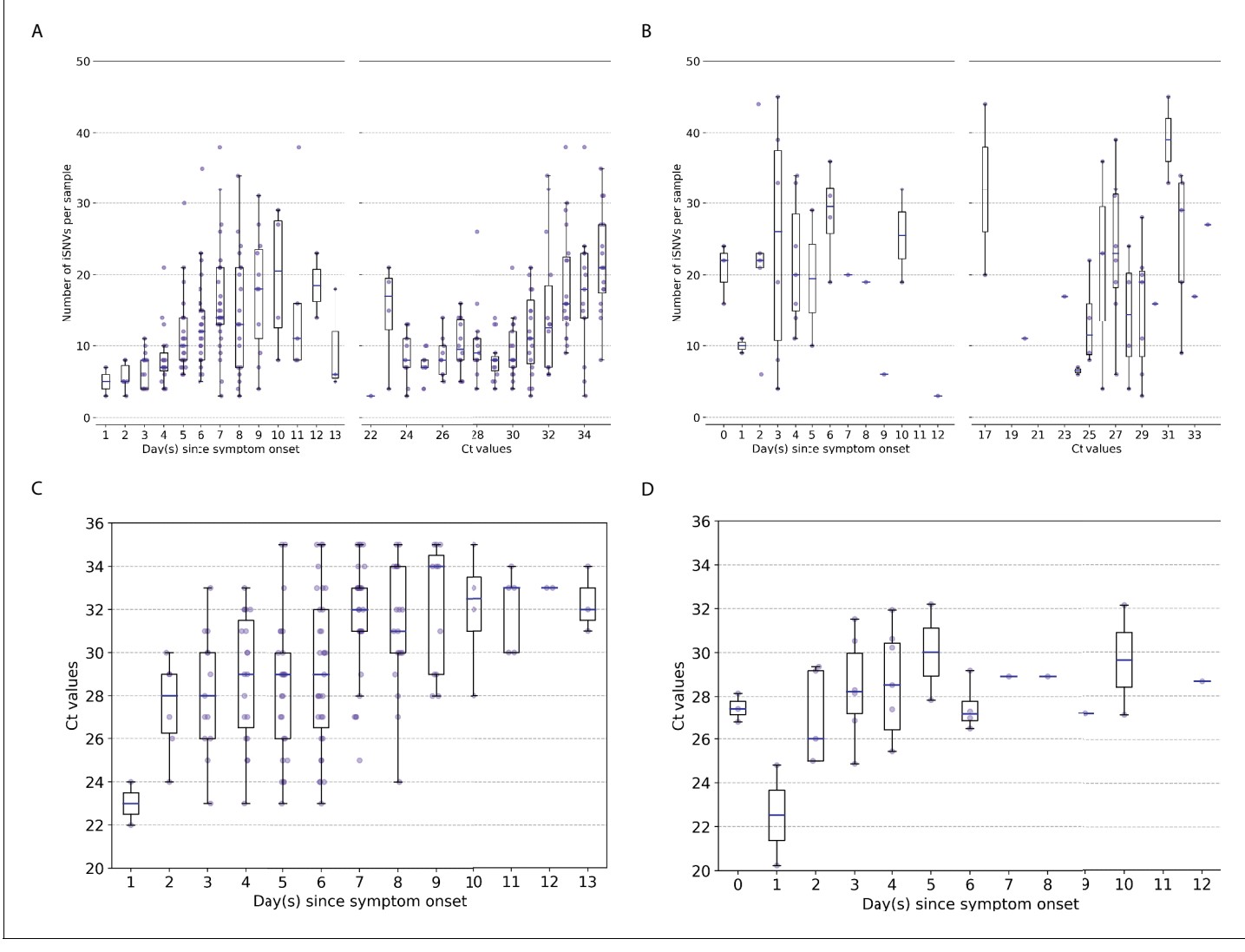

**Figure 1.** Genetic diversity of within-host influenza A virus populations. Box plots summarizing the number of intra-host single-nucleotide variants (iSNVs; median, interquartile range [IQR], and whiskers extending within median ±1.5 × IQR) identified in samples with adequate breadth of coverage across the whole influenza virus genome in (**A**) seasonal A/H3N2 and (**B**) pandemic A/H1N1pdm09 virus samples, stratified by day(s) since symptom onset or qPCR cycle threshold (Ct) values. (**C, D**) Ct values as a function of day(s) since symptom onset for A/H3N2 viruses (**C**) and A/H1N1pdm09 viruses (**D**).

The online version of this article includes the following figure supplement(s) for figure 1:

**Figure supplement 1.** Sequence coverage across all influenza gene segments and samples.

**Figure supplement 2.** Frequencies of nucleotide variants found in A/H3N2 viral reads sequenced from overlapping amplicons.

**Figure supplement 3.** Maximum-likelihood phylogeny of putative majority (consensus) and minority whole-genome sequences (by concatenating all eight gene segments) of A/H3N2 virus samples.

**Figure supplement 4.** Maximum-likelihood phylogeny of putative majority (consensus) and minority whole-genome sequences (by concatenating all eight gene segments) of H1N1pdm09 virus samples.

**Figure supplement 5.** Pearson's correlation between the first day of oseltamivir treatment administered to patients and the last day on which viral samples with cycle threshold (CT) values ≤35 were collected.

treatment (*Koel et al., 2020*), nonsynonymous changes to NP were likely mediated by protein stability, T-cell immune response, and/or host cellular factors (see next section).

For A/H1N1pdm09 viruses during the first wave of the pandemic, the overall within-host evolutionary rate was as high as ~$10^{-4}$ divergence per site per day in some samples on day 0 post-symptom onset (*Figure 2B*). We observed higher nonsynonymous evolutionary rates relative to synonymous ones initially after symptom onset but were unable to determine if they were

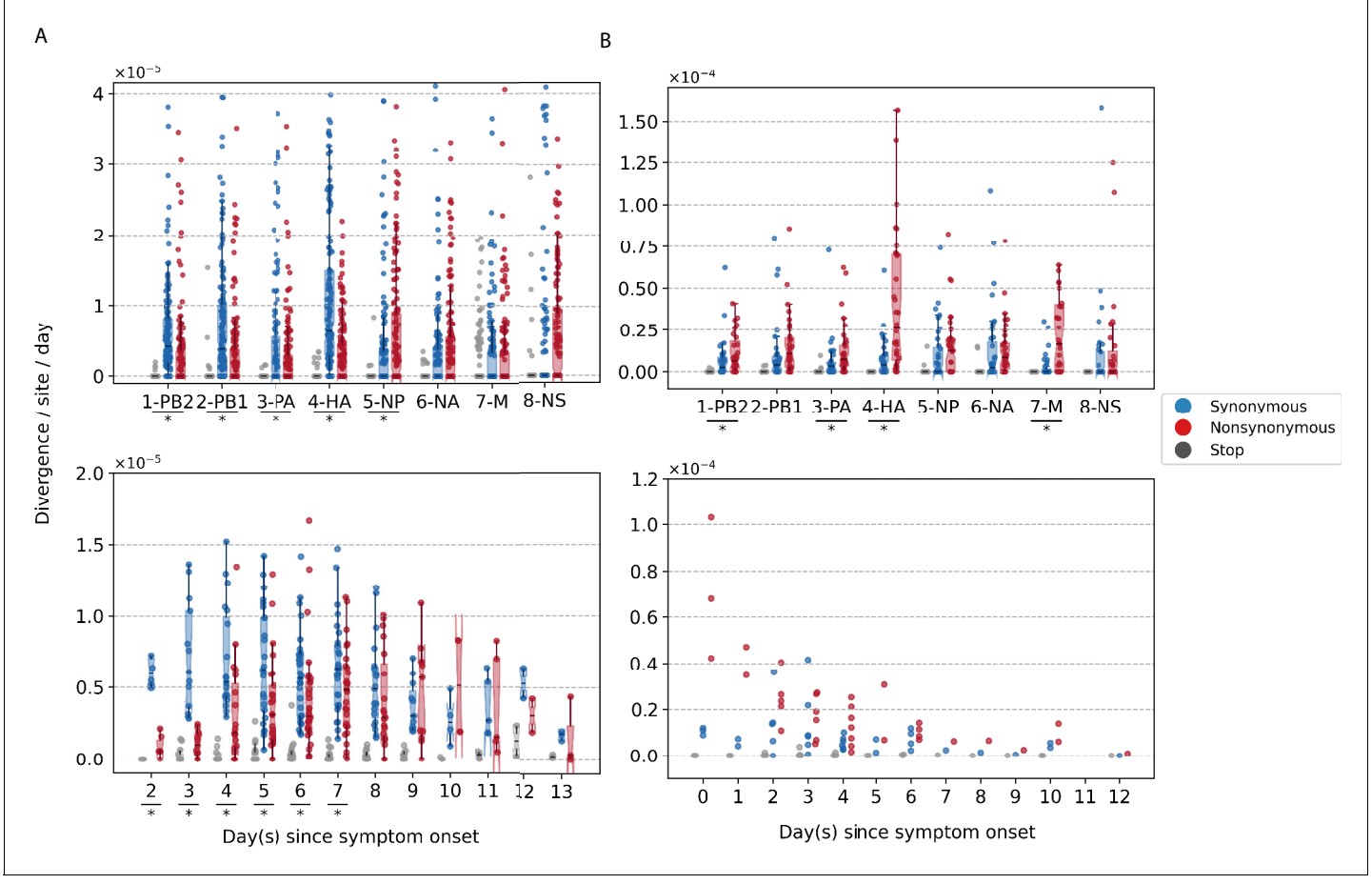

**Figure 2.** Box plots (median, interquartile range [IQR], and whiskers extending within median ±1.5 × IQR) summarizing the empirical within-host evolutionary rates of (A) seasonal A/H3N2 viruses and (B) pandemic A/H1N1pdm09 viruses. Top panel shows the evolutionary rate of individual gene segments over all timepoints ($r_g$) while the bottom panel depicts the genome-wide evolutionary rate ($r_t$) for each day since symptom onset. All rates are stratified by substitution type (synonymous – blue; nonsynonymous – red; gray – stop codon). Wilcoxon signed-rank tests were performed to assess if the paired synonymous and nonsynonymous evolutionary rates are significantly distinct per individual gene segment or timepoint (annotated with * if $p<0.05$). This was done for all sets of nonsynonymous and synonymous rate pairs except for those computed per day since symptom onset for A/H1N1pdm09 viruses due to the low number of data points available (median number of A/H1N1pdm09 virus samples collected per day since symptom onset = 2). Note that the scales of the y axes differ between (A) and (B) to better show rate trends.

The online version of this article includes the following figure supplement(s) for figure 2:

**Figure supplement 1.** Box plots (median, interquartile range [IQR], and whiskers extending within median ±1.5 × IQR) summarizing the empirical within-host evolutionary rates ($r_{g,t}$) of different H3N2 viral gene segments.

**Figure supplement 2.** Box plots (median, interquartile range [IQR], and whiskers extending within median ±1.5 × IQR) summarizing the empirical within-host evolutionary rates ($r_{g,t}$) of different H1N1pdm09 viral gene segments.

**Figure supplement 3.** Linear regression of within-host synonymous and nonsynonymous evolutionary rates of within-host A/H3N2 virus samples.

**Figure supplement 4.** Box plots (median, interquartile range [IQR], and whiskers extending within median ±1.5 × IQR) summarizing the empirical daily within-host evolutionary rates of seasonal A/H3N2 viruses.

significantly different due to the low number of samples (i.e., median = 2 samples per day post-symptom onset). In turn, we also could not meaningfully characterize the temporal trends of within-host evolution for the pandemic virus with this dataset. Nonetheless, consolidating over all samples across all timepoints, there were significantly higher rates of accumulation of nonsynonymous variants in the polymerase basic 2 (PB2), polymerase acidic (PA), HA, and matrix (M) gene segments (*Figure 2B*, *Figure 2—figure supplement 2*, and *Figure 3—figure supplement 2*). All gene segments also yielded NS/S ratios > 1 (*Supplementary file 1*).

## Intra-host minority variants

Most of the iSNVs identified for both virus subtypes were observed at low frequencies (2–5%; *Figure 3*) and appear to be stochastically introduced across the virus genome (*Figure 4*). Purifying selection dominated within-host seasonal A/H3N2 virus populations as the ratio of nonsynonymous to synonymous variants was 0.72 across all samples and variant frequencies (*Figure 3A* and *Figure 3—figure supplement 1*). Of note, the canonical antigenic sites of the HA gene segment (*Wiley et al., 1981*) of the A/H3N2 virus populations experienced strong negative selection as evidenced by the occurrence of synonymous variants (median frequency = 0.14, IQR range = 0.09–0.27) at far greater frequencies relative to those at non-antigenic sites of HA (median frequency = 0.03, IQR range = 0.03–0.05; Mann–Whitney U test $p = 1.18 \times 10^{-24}$; *Figure 4C*). There were no significant differences in the frequencies of nonsynonymous iSNVs between the antigenic sites of H3 (median frequency = 0.04, IQR range = 0.03–0.06) and the rest of the HA gene segment (median frequency = 0.03, IQR range = 0.02–0.06; Mann–Whitney U test $p = 0.29$; *Figure 4C*). In contrast, there were 1.94 times as many nonsynonymous minority iSNVs relative to synonymous ones identified in the pandemic A/H1N1pdm09 virus samples (*Figure 3B* and *Figure 3—figure supplement 2*). Variant frequencies of nonsynonymous iSNVs found in the antigenic epitopes of H1 (*Caton et al., 1982*) (median frequency = 0.04, IQR range = 0.04–0.05) were, however, not significantly different from those of non-antigenic sites (median frequency = 0.05, IQR range = 0.03–0.16; Mann–Whitney U test $p = 0.34$; *Figure 4D*).

As observed in a previous study using different data (*Xue and Bloom, 2020*), premature stop codon (nonsense) mutations accumulated within-host, though only at low rates. Here, we observed similarly low median nonsense rates, ranging between 0 and $1.29 \times 10^{-6}$ divergence per site per day across the entire A/H3N2 virus genome over the course of infection (IQR limits range between 0 and at most, $1.82 \times 10^{-6}$ divergence per site per day; *Figure 2A*). Premature stop codons accumulated in the matrix (M) genes predominantly but also appeared in all other influenza gene segments within various individuals (*Figures 2A* and *4A*). Nonsense mutations also accumulated within the A/H1N1pdm09 virus samples (*Figure 2B*). Similar to A/H3N2 viruses, nonsense

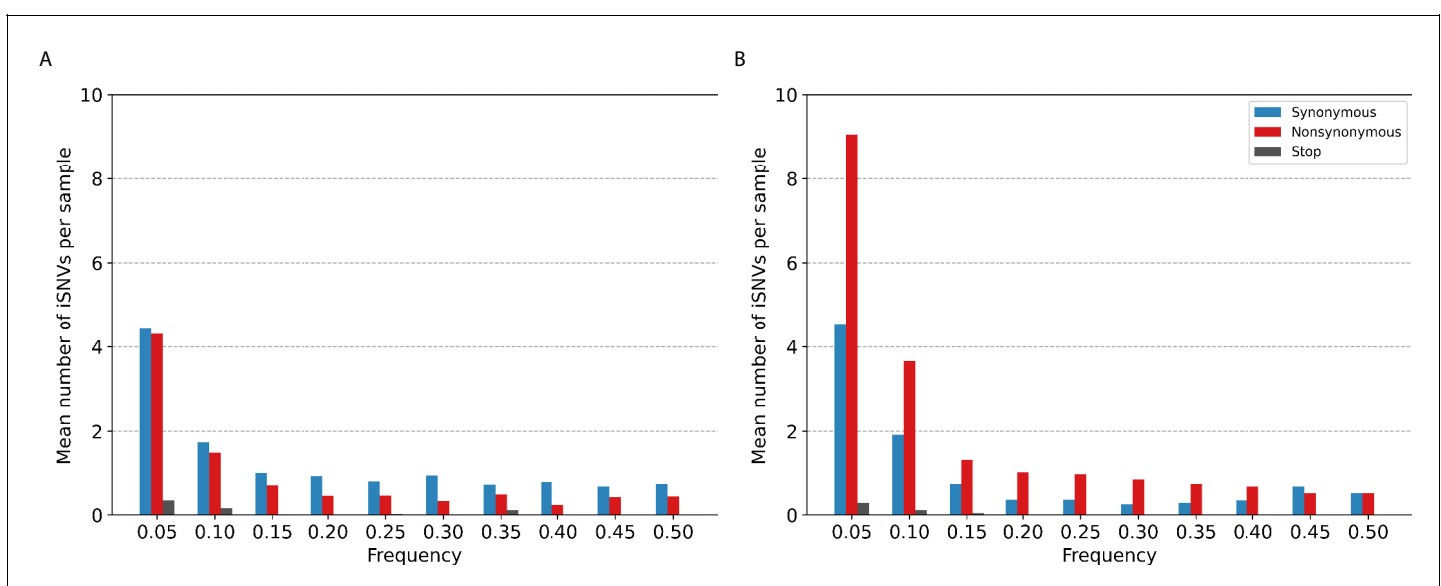

**Figure 3.** Histogram of the mean number of minority intra-host single-nucleotide variants (iSNVs) identified per sample across all. (**A**) A/H3N2 and (**B**) A/H1N1pdm09 virus specimens, sorted by frequency bins of 5% and substitution type (synonymous – blue; nonsynonymous – red; stop-codon – gray). The online version of this article includes the following figure supplement(s) for figure 3:

**Figure supplement 1.** Histogram of the mean number of minority intra-host single-nucleotide variants (iSNVs) identified per sample across all H3N2 viral gene segments across all samples sorted by frequency bins of 5% and substitution type (synonymous – blue; nonsynonymous – red; gray – stop codon).

**Figure supplement 2.** Histogram of the mean number of minority intra-host single-nucleotide variants (iSNVs) identified across all H1N1pdm09 viral gene segments across all samples sorted by frequency bins of 5% and substitution type (synonymous – blue; nonsynonymous – red; stop-codon – gray).

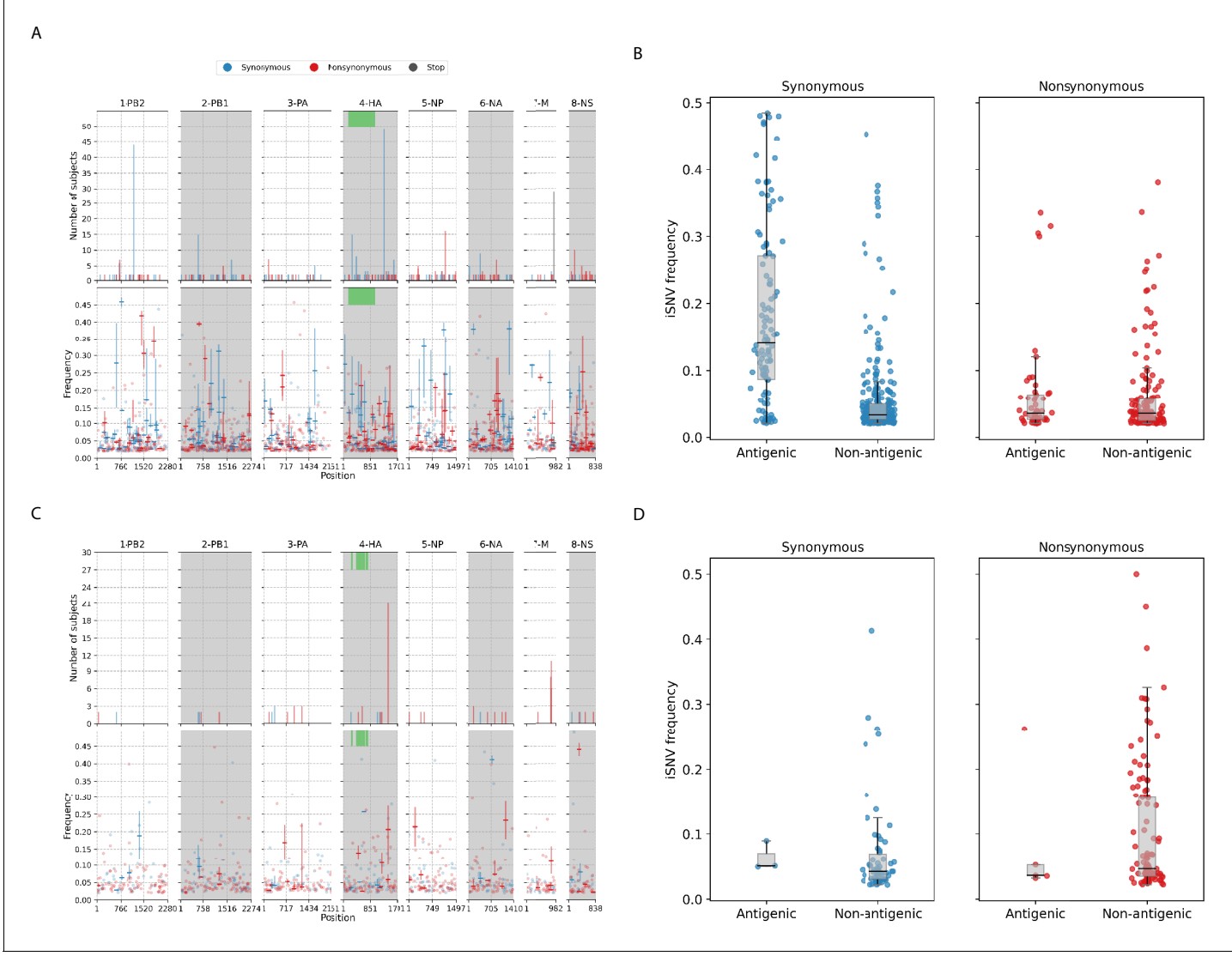

**Figure 4.** Intra-host single-nucleotide variants in within-host IAV populations. (**A**) Breakdown of intra-host single-nucleotide variants (iSNVs) identified in seasonal A/H3N2 virus samples. The top panels plot the nucleotide positions where iSNVs were found in at least two subjects. The bottom panels show the frequencies at which iSNVs were identified. For sites with iSNVs that were found in two or more subjects, the interquartile ranges of variant frequencies are plotted as vertical lines and the median frequencies are marked with a dash. If the iSNV was only found in one subject, its corresponding frequency is plotted as a circle. All iSNVs are stratified to either synonymous (blue), nonsynonymous (red), or stop codon (gray) variants. Only the nonsynonymous variants are plotted if both types of variants are found in a site. Positions of antigenic sites of the hemagglutinin (HA)_ gene segment (*Igarashi et al., 2010*) are marked in green on the top panels. (**B**) Box plots of the frequencies of synonymous and nonsynonymous variants between antigenic and non-antigenic sites of seasonal A/H3N2 HA gene segment. (**C**) Similar plots to (**A**) for iSNVs found in pandemic A/H1N1pdm09 virus samples. (**D**) Similar plots to (**B**) for HA iSNVs identified in the pandemic A/H1N1pdm09 virus samples.

The online version of this article includes the following figure supplement(s) for figure 4:

**Figure supplement 1.** Plots of intra-host hemagglutinin (HA) amino acid variants in A/H3N2-infected individuals.

**Figure supplement 2.** Plots of intra-host neuraminidase (NA) amino acid variants in A/H3N2-infected individuals.

**Figure supplement 3.** Plots of intra-host nucleoprotein (NP) amino acid variants in A/H3N2-infected individuals.

**Figure supplement 4.** Plots of M2 protein intra-host amino acid variants in A/H3N2-infected individuals.

**Figure supplement 5.** Plots of hemagglutinin (HA) intra-host amino acid variants in A/H1N1pdm09-infected individuals.

**Figure supplement 6.** Plots of neuraminidase (NA) intra-host amino acid variants in A/H1N1pdm09-infected individuals.

**Figure supplement 7.** Plots of M2 protein intra-host amino acid variants in A/H1N1pdm09-infected individuals.

**Figure supplement 8.** Frequency distributions of intra-host single-nucleotide variants (iSNVs) below the 2% variant calling threshold found in nucleotide positions NP-1150 and M-917 that encode for amino acid sites NP-384 and M2-77, respectively.

**Figure supplement 9.** Plots of within-host recurring A/H3N2 amino acid variants NP-G384R and M2-R77* based on variant calls and frequencies after remapping sample reads to their respective sample consensus sequence.

mutation rates were much lower compared to the synonymous and nonsynonymous counterparts (median genome-wide rate across all samples between 0 and $1.43 \times 10^{-6}$ divergence per site per day; IQR limits between 0 and $2.18 \times 10^{-6}$ divergence per site per day).

The premature stop codon mutations were mostly found at low frequencies for both influenza subtypes (<10%; *Figure 3*). The exception lies with one of the A/H3N2 virus samples where a premature stop codon was found in position 77 of the M2 ion channel with variant frequency as high as 34.6% (patient 1843, day 6 since symptom onset; *Figure 4A* and *Figure 4—figure supplement 4*). The premature stop codon in M2-77 was also found in 27 other individuals across multiple timepoints, albeit at a much lower frequency that never amounted more than 10% (*Figure 4A* and *Figure 4—figure supplement 4*). This was unlikely to be a sequencing artifact resulting from a mistaken incorporation of the primer sequence as its carboxyl terminal falls outside the coding region of the M gene segment (*Supplementary file 3*) and the variant frequencies would have been much higher in all samples if this was the case.

Despite the dominance of purifying selection in seasonal A/H3N2 intra-host viral populations, we detected several nonsynonymous variants of interest. Amino acid variants emerging in the HA and NA proteins were discussed in a previous work (*Koel et al., 2020*; Appendix A1). In the NP, there were two notable nonsynonymous variants, D101N/G and G384R, that appeared in multiple individuals who were sampled independently between 2007 and 2009 (*Figure 4A* and *Figure 4—figure supplement 3*). D101N/G was found in seven different patients and at least for D101G the mutation was previously linked to facilitating escape from MxA, a key human antiviral protein (*Mänz et al., 2013*). However, the nonsynonymous mutation was only found in low frequencies and remained invariant during the respective courses of infection for all seven patients (median variant frequency across all samples = 0.03; IQR = 0.02–0.07).

NP-G384R emerged in 16 unlinked patients infected by A/H3N2 virus. Even though G384R did not become the majority variant in any of these individuals (median variant frequency across all samples = 0.14; IQR = 0.07–0.20), the variant emerged around days 4–5 post-symptom onset and mostly persisted within each individual for the rest of sampled timepoints. G384R is a stabilizing mutation in the A/Brisbane/10/2007 A/H3N2 virus NP background (*Ashenberg et al., 2013*) that is similar to the viruses investigated here. Interestingly, position 384 is an anchor residue for several NP-specific epitopes recognized by specific cytotoxic T lymphocytes (CTLs) that are under continual selective pressure for CTL escape (*Berkhoff et al., 2005*; *Gog et al., 2003*). The wild-type glycine residue is known to be highly deleterious even though it was shown to confer CTL escape among HLA-B*2705-positive individuals (*Gong et al., 2013*; *Rimmelzwaan et al., 2004*; *Berkhoff et al., 2004*).

Using a maximum-likelihood (ML) approach to reconstruct and estimate the frequencies of the most parsimonious haplotypes of each gene segment, we computed linkage disequilibrium and found evidence of potential epistatic co-variants to NP-G384R in the A/H3N2 virus populations of multiple individuals (*Figure 5* and *Supplementary file 2*). When analyzing how these variants could alter protein stability using FoldX, the stabilizing effects of G384R (mean $\Delta\Delta G = -3.84$ kcal/mol [SD = 0.06 kcal/mol]) were found to alleviate the likely destabilizing phenotype of a functionally relevant linked variant in two of the three co-mutation pairs identified in separate individuals (i.e., G384R/M426I and G384R/G102R; *Supplementary file 2*). In the first individual (subject 1224), M426I was inferred to have emerged among the viral haplotypes encoding NP-G384R on the 10th day post-symptom onset (D10). M426I may be compensating for T-cell escape that was previously conferred by 384G even though the two amino acid sites are anchor residues of different NP-specific CTL epitopes (*Berkhoff et al., 2005*). M426I was found to be highly destabilizing (mean $\Delta\Delta G = 2.61$ kcal/mol [standard deviation, SD = 0.05 kcal/mol]; *Table 1*) but when co-mutated with G384R, stability changes to NP were predicted to be neutral (mean $\Delta\Delta G = -0.42$ kcal/mol [SD = 0.06 kcal/mol]). In the second individual (subject 1686), G102R was likely linked to G384R in the within-host virus populations found in the D10 sample. As a single mutant, G102R is also destabilizing to NP (mean $\Delta\Delta G = 4.87$ kcal/mol [SD = 0.00 kcal/mol]). However, when combined with G384R, NP stability was only weakly destabilizing (mean $\Delta\Delta G = 0.76$ kcal/mol [SD = 0.09 kcal/mol]). G102R was previously found to bypass the need for cellular factor importin-$\alpha$7, which is crucial for viral replication and pathogenicity of IAVs in humans (*Resa-Infante et al., 2015*; *Resa-Infante et al., 2019*; *Gabriel et al., 2011*).

For the pandemic A/H1N1pdm09 viruses, most of the nonsynonymous variants were found singularly in individual patients (*Figure 4B*). Putative HA antigenic minority variants were found in four

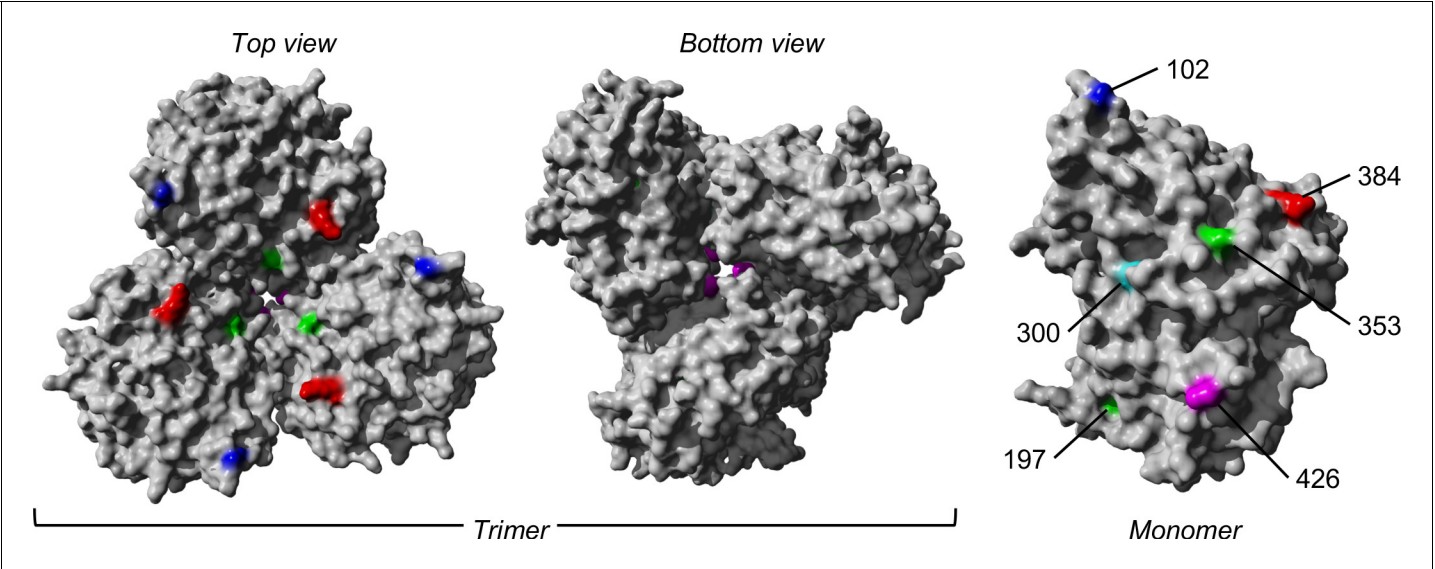

**Figure 5.** The trimeric and monomeric crystal structures of nucleoprotein (PDB: 3ZDP) (*Chenavas et al., 2013*) of influenza A viruses. Amino acid sites with potentially linked epistatic amino acid variants as tabulated in *Table 1* are separately colored, with their corresponding positions annotated on the monomeric structure.

individuals in distinct amino acid sites (G143E, N159K, N197K, and G225D; H3 numbering without signal peptide; *Figure 4—figure supplement 5*). All of these variants were found at frequencies $\leq 5\%$ and the wild-type residues have been conserved in the corresponding positions globally to date, with the exception of position 225. Here, HA-225G was the majority variant (76%) in a hospitalized individual (subject 11-1022; *Supplementary file 4*) and D225G is linked to infections with severe disease outcomes (*Mak et al., 2010*). Furthermore, one of the few nonsynonymous iSNVs that co-emerged in multiple unlinked patients was found in the usually conserved stem of the HA protein, L455F/I (H3 numbering without signal peptide), appearing in 17 separate individuals (*Figure 4B* and *Figure 4—figure supplement 5*). The amino acid variant was found in patients from different time periods and geographical locations (*Supplementary file 4*), thus it is unlikely that this was a unique

**Table 1.** FoldX stability predictions of likely linked nonsynonymous minority variants found in A/H3N2 nucleoprotein.

The mean $\Delta\Delta G$ and standard deviation (SD) values reported are based on the results of five distinct simulations. Variants with mean $\Delta\Delta G < -0.46$ kcal/mol are deemed to be stabilizing while destabilizing mutants were estimated to yield $\Delta\Delta G > 0.46$ kcal/mol.

| | $\Delta\Delta G$ **(kcal/mol)** | |
| --- | --- | --- |
| **Variants** | **Mean** | **SD** |
| G384R | −3.84 | 0.06 |
| M426I | 2.61 | 0.05 |
| G384R,M426I | −0.42 | 0.06 |
| G102R | 4.87 | 0.00 |
| G384R,G102R | 0.76 | 0.09 |
| A493T | 11.96 | 0.30 |
| G384R,A493T | 5.56 | 0.19 |
| V197I | −3.11 | 0.02 |
| S353Y | −1.97 | 0.68 |
| V197I,S353Y | −4.48 | 0.14 |

variant shared among individuals in the same transmission cluster. It was observed as early as day 0 post-symptom onset for some patients and seemed to persist during the infection but only as a minority variant at varying frequencies (median frequency across all samples with mutation = 0.20; IQR = 0.08–0.28). However, this position has also been conserved with the wild-type Leucine residue in the global virus population to date. Hence, it is unclear if HA-L455F/I actually confers any selective benefit even though it was independently found in multiple patients.

We also found oseltamivir resistance mutation H275Y (*Mai et al., 2010*) in the NA proteins in two unlinked individuals who were infected with the A/H1N1pdm09 virus and treated with oseltamivir (*Figure 4—figure supplement 6* and *Supplementary file 4*). 275Y quickly became the majority variant in both patients within 3–4 days after the antiviral drug was first administered. Finally, there were two other amino acid variants in the M2 ion channel that appeared within multiple subjects in parallel across different geographical locations – L46P and F48S were identified in 8 and 16 patients, respectively, in a range of frequencies (L46P: median frequency = 0.04, IQR = 0.04–0.05; F48S: median frequency = 0.08, IQR = 0.03–0.13) but similarly, never becoming a majority variant in any of them (*Figure 4* and *Figure 4—figure supplement 7*). Again, the wild-type residues were mostly conserved in the global virus population since the pandemic.

## Within-host simulations

To investigate the evolutionary pressures that likely underpin the observed within-host dynamics of A/H3N2 viruses in young children (*Figure 2*), we performed forward-time Monte Carlo simulations. Given that the median age of the children infected by A/H3N2 virus at the time of sample collection was 2 years of age (IQR = 2–3 years), most of them were likely experiencing one of their first influenza virus infections. Furthermore, influenza vaccination for children is not part of the national vaccination program in Vietnam. As such, most of the children analyzed here lacked influenza virus-specific antibodies based on hemagglutination inhibition assays (*Koel et al., 2020*). Since seasonal A/H3N2 viruses have circulated within the human population since 1968, the virus is well adapted to human hosts at this point such that most nonsynonymous mutations are likely highly deleterious and would not reach detectable frequencies. We hypothesized that detected variants are mostly expected to be weakly deleterious, and thus not purged fast enough by selection such that mutation-selection balance was observed.

Our simulations used a simple within-host evolution model represented by a binary genome that distinguishes between synonymous and nonsynonymous loci. Given that the estimated transmission bottleneck sizes for seasonal A/H3N2 viruses (*McCrone et al., 2018*; *Ghafari et al., 2020*) are narrow at 1–2 genomes, we modeled an expanding virus population size during the initial timepoints of the infection that started with one virion. If within-host virus populations were to evolve neutrally, we would observe similar synonymous and nonsynonymous evolutionary rates throughout the infection (*Figure 6A*). On the other hand, if negative selection is sufficiently strong, accumulation of deleterious nonsynonymous variants will decrease substantially with time (*Figure 6B*). Clearly, these patterns were not observed for A/H3N2 viruses (*Figure 2A*). However, if most de novo nonsynonymous mutations are only weakly deleterious, we would observe larger synonymous evolutionary rates initially before nonsynonymous variants accumulate to similar levels (*Figure 6C*). By then, virion population size ($N$) would also be large enough relative to the virus mutation rate (μ) (i.e., $N\mu \gg 1$; Appendix A5) such that mutation-selection balance is expected and evolutionary rates remain fairly constant, similar to the patterns empirically observed for within-host A/H3N2 virus populations (*Figure 2A*).

## Discussion

Multiple next-generation sequencing studies have found little evidence of positive selection in seasonal influenza virus populations of acutely infected individuals (*McCrone et al., 2018*; *Dinis et al., 2016*; *Debbink et al., 2017*; *Valesano et al., 2020*; *Sobel Leonard et al., 2016*; *McCrone et al., 2020*). Recent modeling work showed that the time required to initiate new antibody production and asynchrony with virus exponential growth limits the selection of de novo antigenic variants within host in acute seasonal influenza virus infections (*Morris et al., 2020*). In contrast, phenotypically relevant variants that were positively selected in within-host virus populations of severely immunocompromised patients coincided with those selected by the global seasonal IAV population (*Xue et al., 2017*; *Lumby et al., 2020*). This implies that within-host evolutionary dynamics of seasonal IAVs in

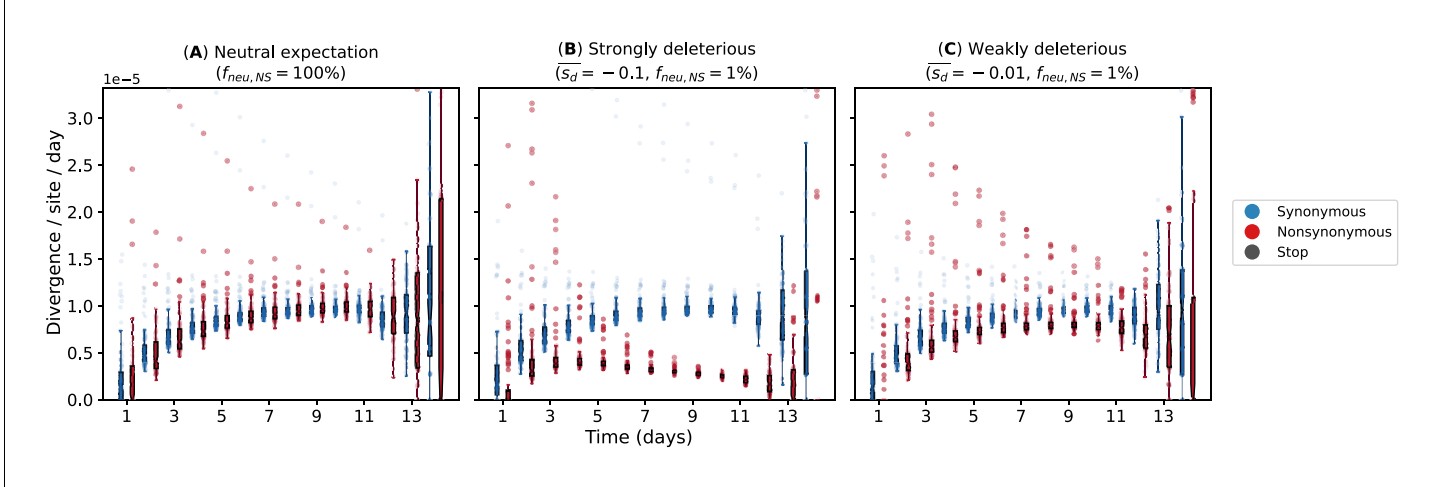

**Figure 6.** Evolutionary rates computed from forward-time Monte Carlo within-host simulations for different mean deleterious effects ($\bar{s}_d$) of nonsynonymous mutations. We assumed that synonymous mutations are neutral for all simulations. (**A**) Neutral expectation where all nonsynonymous mutations are neutral ($f_{neu,NS} = 100\%$). We tested our hypotheses where the majority of nonsynonymous mutations are non-neutral ($f_{neu,NS} = 1\%$) and they are either (**B**) strongly ($\bar{s}_d = -0.1$) or (**C**) weakly ($\bar{s}_d = -0.01$) deleterious.

The online version of this article includes the following figure supplement(s) for figure 6:

**Figure supplement 1.** Number of virions ($N$) against replicative generation ($t$) based on a target cell-limited within-host model.

immunocompromised individuals are likely to be substantially different owing to the increased time for virus diversity to accumulate and for selection to act (*Petrova and Russell, 2018*). In other words, the duration of infection is likely to be critical for positive evolutionary selection to be effective within host.

Viral shedding duration is often longer in young children infected with seasonal influenza virus compared to otherwise healthy adults (*Ng et al., 2016*). Children also play a critical role in 'driving' influenza epidemics due to their higher contact and transmission rates (*Worby et al., 2015*; *Viboud et al., 2004*). As such, our seasonal A/H3N2 virus results fill an important gap in the current literature of within-host evolutionary studies of seasonal IAVs as most of the samples analyzed were collected from children under the age of 6 years up to 2 weeks post-symptom onset. Importantly, the absence of antibody-mediated immunity in young unvaccinated children, which would otherwise reduce the extended duration of infection, has the potential to facilitate other routes of virus evolution.

Similar to the aforementioned within-host studies, the A/H3N2 virus population within these children was characterized by low genetic diversity and dominated by purifying selection early in the infection. Due to a lack of antibody response against the antigenic regions of HA (*Koel et al., 2020*), it is unsurprising that we observed a lack of adaptive changes to the HA antigenic regions, similar to adults in previous studies (*McCrone et al., 2018*). We also found that the polymerase genes were subjected to purifying selection, indicating their critical role in virus replication as negative selection purges deleterious variation. However, while purifying selection is detectable, it is incomplete (*Xue and Bloom, 2020*). We observed that most nonsynonymous variants began to accumulate around 3–4 days post-symptom onset, with incrementally higher empirical rates as the infection progressed.

Through simulations of a within-host evolution model, we investigated the hypothesis that in the absence of any positive selection, the accumulation of nonsynonymous iSNVs was a result of their neutral or only weakly deleterious effects and the expanding within-host virion population size during later timepoints in longer infections of naïve young children such that mutation-selection balance was reached. In contrast, this balance was not detected in otherwise healthy older children or adults with short-lived influenza virus infections lasting no more than a week where de novo nonsynonymous iSNVs are rarely found (*McCrone et al., 2018*; *Dinis et al., 2016*; *Debbink et al., 2017*; *Valesano et al., 2020*; *Sobel Leonard et al., 2016*; *McCrone et al., 2020*). The maintenance of genetic diversity through mutation-selection balance within these young children may provide

opportunities for the emergence of phenotypically relevant mutations, which deleterious effects could be alleviated by the accumulation of a secondary compensatory mutations. For example, in one individual NP-G384R was accompanied by NP-M426I, which is an anchor residue of a CTL epitope of NP, abrogating recognition by HLA-B*3501-positive CTLs (*Berkhoff et al., 2005*) but is likely to be deleterious based on our computational protein stability predictions. G384R, which is located in a CTL epitope distinct from M426I (*Berkhoff et al., 2005*), was previously shown to be a stabilizing substitution (*Ashenberg et al., 2013*).

Interestingly, we also observed G384R in the minority virus population of 15 other unlinked individuals. Besides improving NP stability, G384R restores recognition by HLA-B*2705-positive NP-specific CTLs (*Berkhoff et al., 2004*). The NP gene segment in the global A/H3N2 virus population has an evolutionary history of fixating destabilizing amino acid mutations that promote CTL immune escape alongside stabilizing substitutions that compensate for the deleterious effects of the former (*Gong et al., 2013*). The reversal R384G mutation confers CTL escape but is known to be highly deleterious. This substitution was fixed in the global A/H3N2 virus population during the early 1990s as other substitutions such as S259L and E375G epistatically alleviated its destabilizing effects (*Gong et al., 2013*). One possible explanation for the emergence of G384R as a minority variant within these unlinked individuals is that they are all HLA-B*2705 negative. However, we did not collect the necessary blood samples to investigate this possibility.

In contrast, we found a substantially higher fraction of nonsynonymous variants in the within-host virus populations of individuals infected with A/H1N1pdm09 virus during the pandemic. Owing to the low number of A/H1N1pdm09 virus samples and different next-generation sequencing platforms used to sequence samples of the two virus subtypes and consequently differences in base calling error rates and depth of coverage (*Figure 1—figure supplement 1*), we were unable to directly compare the observed levels of within-host genetic diversity and its temporal trends between the two influenza subtypes here. However, given that only iSNVs with frequencies $\geq 2\%$ were called, low-frequency minority variants arising from technical-related errors should be minimized (*Watson et al., 2013*). Importantly, the relative number of nonsynonymous iSNVs identified was far greater than synonymous ones in the pandemic A/H1N1pdm09 virus infections, suggesting that there was room for further human host adaptation, particularly in the HA but also in the polymerase gene segments similar to those observed in other zoonotic influenza virus infections (*Welkers et al., 2019*).

Given the tight estimated transmission bottleneck size (Appendix A4), the relatively large number of iSNVs identified at the start of symptom onset and simulations of within-host evolution (*Figures 1B*, *2B,* and *6D*), it is unlikely that the initial within-host A/H1N1pdm09 virus populations sampled were the inoculating population that founded the infection. Instead, the inoculating viral population had already undergone substantial within-host replication during the incubation period before symptom onset. In fact, four of the individuals analyzed were asymptomatic (i.e., H058/S02, H089/S04, H186/S05, and H296/S04; *Supplementary file 4*). Additionally, pre-symptomatic virus shedding was observed in some of the secondary household cases (*Thai et al., 2014*) and pre-symptomatic transmission has been documented in other settings (*Suess et al., 2012*). Nonetheless, this would not meaningfully impact our conclusions as most of the within-host viral populations sampled at the start of symptom onset should still constitute those found early in infection and the contrasting feature where nonsynonymous iSNVs outnumbered synonymous ones were not observed in the seasonal A/H3N2 virus samples.

For both A/H3N2 and A/H1N1pdm09 virus samples, nonsense iSNVs resulting in premature stop codons were found to accumulate within host, even though only at low proportions. The accumulation of premature stop codon mutations further suggests that while purifying selection dominates within-host influenza virus populations, it may not be acting strongly enough to completely purge these lethal nonsense mutations (*Xue and Bloom, 2020*). Additionally, it has been recently found that incomplete influenza virus genomes frequently occur at the cellular level and that efficient infection depends on the complementation between different incomplete genomes (*Jacobs et al., 2019*). As such, nonsense mutations may not be as uncommon as previously thought. In particular, nonsense mutations in position 77 of the M2 ion channel were independently found in 27 unlinked individuals infected by A/H3N2 virus. While these nonsense mutations are generally considered to be lethal, ion channel activity is retained even if the M2 protein was prematurely truncated up to position 70 at its cytoplasmic tail (*McCown and Pekosz, 2005*).

Our study has several limitations. The number of iSNVs identified can potentially be biased by variations in sequencing coverage (*Zhao and Illingworth, 2019*). As such, the number of iSNVs observed in one intra-host virus populations may not be directly comparable to another with a distinct coverage profile (*Figure 1—figure supplement 1*). As an alternative, the nucleotide diversity $\pi$ statistic (*Nei and Li, 1979*) may be a more robust measure of within-host diversity as it solely depends on the underlying variant frequencies (*Zhao and Illingworth, 2019*). Computing the corresponding $\pi$ statistics for our data, we observed trends in genetic diversity that were similar to those inferred using iSNV counts (Appendix A2 and *Appendix 1—figure 1*).

To ensure accurate measurements of virus diversity in intra-host populations, we would also need to be certain that the estimated variant frequencies precisely reflect the distributions of variants that comprise the sampled virus populations. The inferred variant frequencies can be significantly distorted if virus load is low (*Illingworth et al., 2017*; *Xue et al., 2018*). As such, we limited our analyses for both virus subtypes to samples with Ct values ≤35, which likely afford sufficient virus material for sequencing (*Xue et al., 2018*). We were unable to estimate the amount of frequency estimation errors for the A/H1N1pdm09 virus samples as only one sequencing replicate was performed using the universal 8-segment PCR method (*Hoffmann et al., 2001*). However, for the A/H3N2 virus samples, independent PCR reactions were performed using three partly overlapping amplicons for all gene segments other than the nonstructural and matrix genes. We compared the variant frequencies estimated for any overlapping sites generated by reads derived from distinct amplicons with sufficient coverage (>100×). Variant frequencies computed from independent amplicons agreed well with each other across the range of Ct values of the samples from which variants were identified (*Figure 1—figure supplement 2*), affirming the precision of our iSNV frequency estimates for the A/H3N2 virus samples, including those with higher Ct values.

We also performed additional checks to ensure that our results were not driven by potential PCR and/or technical artifacts. First, we excluded all iSNVs found under the 75th percentile of frequency range of A/H3N2 variants that were found in only one of the overlapping amplicons. We then recomputed the daily within-host evolutionary rates with the remaining iSNVs (*Figure 2—figure supplement 3*) and found that the relative temporal trends in synonymous and nonsynonymous rates remain similar to those in *Figure 2A*. We also checked that the distributions of frequencies for iSNVs found in recurrent mutation sites (i.e., NP-384 and M2-77) that are below variant calling threshold are comparable to those found in their neighboring sites (±10 nucleotide positions; *Figure 4—figure supplement 8*). Furthermore, we remapped the sample reads to their respective consensus sequences to minimize mapping of technical artifacts. We were still able to detect the recurring NP-G384R and M2-R77* amino acid mutations in multiple individuals and timepoints at similar frequencies when mapped to the reference genome (*Figure 4—figure supplements 4*, *5,* and *9*). As such, these recurrent mutations are unlikely to have been resulted from erroneous variant calls of artifacts.

Finally, most study participants received oseltamivir during the course of their infections (*Supplementary file 4*). Although we were unable to identify any potential effects of enhanced viral clearance or any other evolutionary effects due to the treatment, besides oseltamivir resistance-associated mutations, it is unlikely that the antiviral treatment had a substantial impact on our results. First, the median timepoint in which the antiviral was initially administered was 4 days post-symptom onset (IQR = 3–6 days; *Supplementary file 4*). Previous studies showed that enhanced viral clearance of IAVs was mostly observed among patients who were treated with oseltamivir within 3 days of symptom onset (*South East Asia Infectious Disease Clinical Research Network, 2013*; *Lee et al., 2009*; *Ling et al., 2010*). Of note, late timepoint samples in this study (≥8 days since symptom onset) mostly came from individuals who started oseltamivir treatments ≥4 days post-symptom onset (*Figure 1—figure supplement 5*). Second, at least in vitro, there were no differences in the levels of genetic diversity observed in influenza virus populations after multiple serial passages whether they were treated with oseltamivir or not (*Renzette et al., 2014*).

To conclude, we presented how intra-host populations of seasonal and pandemic influenza viruses are subjected to contrasting evolutionary selection pressures. In particular, we showed that the evolutionary dynamics and ensuing genetic variation of these within-host virus populations changes during the course of infection, highlighting the importance for sequential sampling, particularly for longer-than-average infections such as those in the young children studied here.

## Materials and methods

### Sample collection and viral sequencing

The A/H3N2 virus samples were collected from 52 patients between August 2007 and September 2009 as part of an oseltamivir dosage trial conducted by the South East Asia Infectious Disease Clinical Research Network (SEAICRN), which is detailed in a previous work (*South East Asia Infectious Disease Clinical Research Network, 2013*). Briefly, patients with laboratory-confirmed influenza virus infection and duration of symptoms ≤10 days were swabbed for nose and throat samples daily between 0 and 10 days as well as day 14 upon enrolment for the study (*Supplementary file 4*). All PCR-confirmed A/H3N2 virus samples with cycle threshold (Ct) values ≤35 were included for sequencing.

Library preparation and viral sequencing protocols performed on these A/H3N2 virus samples are elaborated in detail in *Koel et al., 2020*. Here, we highlight the key aspects of our preparation and sequencing procedures. Using segment-specific primers (*Supplementary file 3*), we performed six independent PCR reactions, resulting in three partly overlapping amplicons for each influenza virus gene segment other than the matrix (M) and nonstructural (NS) genes where a single amplicon was produced to cover the entirety of the relatively shorter M and NS genes. The use of shorter but overlapping amplicons in the longer gene segments improves amplification efficiency, ensuring that these longer segments are sufficiently covered should there be any RNA degradation in the clinical specimen. These overlapping PCR products were pooled in equimolar concentrations for each sample and purified for subsequent library preparation. Sequencing libraries were prepared using the Nextera XT DNA Library Preparation kit (Illumina, FC-131-1096) as described in *Koel et al., 2020*. Library pools were sequenced using the Illumina MiSeq 600-cycle MiSeq Reagent Kit v3 (Illumina, MS-102-3003).

The A/H1N1pdm09 virus samples were obtained as part of a household-based influenza virus cohort study that was also performed by SEAICRN. The study was conducted between July and December 2009, involving a total of 270 households in Ha Nam province, Vietnam (*Horby et al., 2012*). Similarly, combined nose and throat swabs were collected daily for 10–15 days from individuals with influenza-like illness (i.e., presenting symptoms of fever >38°C and cough, or sore throat) and their household members, including asymptomatic individuals (*Supplementary file 4*). We also analyzed additional samples collected from unlinked hospitalized patients who were infected by the A/H1N1pdm09 virus from two major Vietnamese cities (Hanoi and Ho Chih Minh) during the first wave of the pandemic (*South East Asia Infectious Disease Clinical Research Network, 2013*; *Hien et al., 2010*). A total of 32 PCR-confirmed A/H1N1pdm09-infected individuals originating from both households and hospitalized cases were selected for sequencing based on availability and Ct values ≤33 (*Supplementary file 4*).

For the A/H1N1pdm09 virus samples, RNA extraction was performed manually using the High Pure RNA isolation kit (Roche) with an on-column DNase treatment according to the manufacturer's protocol. Total RNA was eluted in a volume of 50 μl. Universal influenza virus full-genome amplification was performed using a universal 8-segment PCR method as described previously (*Watson et al., 2013*; *Zhou et al., 2009*; *Jonges et al., 2014*). In short, two separate RT-PCRs were performed for each sample, using primers common-uni12R (5′-GCCGGAGCTCTGCAGAT ATCAGC RAAAGCAGG-3′), common-uni12G (5′-GCCGGAGCTCTG CAGATATCAGCGAAAGCAGG-3′), and common-uni13 (5′-CAGGAA ACAGCTATGACAGTAGAAACAAGG-3′). The first RT-PCR mixture contained the primers common-uni12R and common-uni13. The second RT-PCR mixture contained the primers common-uni12G and common-uni13, which greatly improved the amplification of the PB2, PB1, and PA segments. Reactions were performed using the One-Step RT-PCR kit High Fidelity (Invitrogen) in a volume of 50 μl containing 5.0 μl eluted RNA with final concentrations of 1xSuperScript III One-Step RT-PCR buffer, 0.2 μM of each primer, and 1.0 μl SuperScript III RT/Platinum Taq High Fidelity Enzyme Mix (Invitrogen). Thermal cycling conditions were as follows: reverse transcription at 42°C for 15 min, 55°C for 15 min, and 60°C for 5 min; initial denaturation/enzyme activation of 94°C for 2 min; 5 cycles of 94°C for 30 s, 45°C for 30 s, slow ramp (0.5 °C/s) to 68°C, and 68°C for 3 min; 30 cycles of 94°C for 30 s, 57°C for 30 s, and 68°C for 3 min; and a final extension of 68°C for 5 min. After the PCR, equal volumes of the two reaction mixtures were combined to produce a well-distributed mixture of all eight influenza virus segments. All RT-PCRs were performed in duplicate. Samples were diluted to a DNA concentration of 50 ng/μl followed by ligation of 454 sequencing

adaptors and molecular identifier (MID) tags using the SPRIworks Fragment Library System II for Roche GS FLX+ DNA Sequencer (Beckman Coulter), excluding fragments smaller than 350 base pairs, according to the manufacturer's protocol to allow for multiplex sequencing per region. The quantity of properly ligated fragments was determined based on the incorporation efficiency of the fluorescent primers using FLUOstar OPTIMA (BMG Labtech). Emulsion PCR, bead recovery and enrichment were performed manually according to the manufacturer's protocol (Roche), and samples were sequenced in Roche FLX+ 454. Sequencing was performed at the Sanger Institute, Hinxton, Cambridge, England, as part of the FP7 program EMPERIE. Standard flowgram format (sff) files containing the filter passed reads were demultiplexed based on the molecular identifier (MID) sequences using QUASR package version 7.0 (*Watson et al., 2013*).

## Read mapping

Trimmomatic (v0.39; *Bolger et al., 2014*) was used to discard reads with length <30 bases while trimming the ends of reads where base quality scores fall below 20. The MAXINFO option was used to perform adaptive quality trimming, balancing the trade-off between longer read length and tolerance of base calling errors (target length = 40, strictness = 0.4). For the A/H3N2 virus samples, the trimmed paired reads were merged using FLASH (v1.2.11) (*Magoč and Salzberg, 2011*). All remaining reads were then locally aligned to A/Brisbane/10/2007 genome (GISAID accession: EPI_ISL_103644) for A/H3N2 virus samples and A/California/4/2009 genome (EPI_ISL_376192) for A/H1N1pdm09 virus samples using Bowtie2 (v2.3.5.1) (*Langmead and Salzberg, 2012*). Aligned reads with mapping scores falling below 20 alongside bases with quality score (*Q-score*) below 20 were discarded.

## Variant calling and quality filters

Minority variants of each nucleotide site with a frequency of at least 2% were called if the nucleotide position was covered at least 50× (H1N1pdm09) or 100× (H3N2) and the probability that the variant was called as a result of base calling errors ($p_{Err}$) was less than 1%. $p_{Err}$ was modeled by binomial trials (*Illingworth, 2016*):

$$p_{Err} = \sum_{i=n}^{N} \binom{N}{i} p_e^i (1 - p_e)^{N-i}$$

where $p_e = -10^{-\frac{Q-score}{10}}$, $N$ is the coverage of the nucleotide site in question, and $n$ is the absolute count of the variant base tallied.

While lower coverage at both ends of individual gene segments was expected, there were also variable coverage results across gene segments for some samples that were mapped to A/H3N2 virus (*Figure 1—figure supplement 1*). In order to retain as many samples deemed to have adequate coverage across whole genome, a list of polymorphic nucleotide sites found to have >2% minority variants in more than one sample was compiled. Each gene segment of a sample was determined to achieve satisfactory coverage if >70% of these polymorphic sites were covered at least 100×. For A/H1N1pdm09, the gene segment of a sample was deemed to be adequately covered if 80% of the gene was covered at least 50×.

The number of iSNVs observed in A/H3N2 virus samples collected from subject 1673 (39–94 iSNVs in three samples collected from 3 [D3] to 5 [D5] days post-symptom onset) and the D8 sample for subject 1878 (73 iSNVs) were substantially greater than the numbers in all other samples. The putative majority and minority segment-concatenated sequences of these samples did not cluster as a monophyletic clade among themselves phylogenetically (*Figure 1—figure supplement 3*), suggesting that these samples might be the product of mixed infections or cross-contamination. These samples were consequently excluded from further analyses.

## Empirical within-host evolutionary rate

The empirical within-host evolutionary rate ($r_{g,t}$) of each gene segment ($g$) in a sample collected on $t$ day(s) since symptom onset was estimated by

$$r_{g,t} = \frac{\sum_i^{n_{g,t}} f_{g,t,i}}{n_{g,t} \cdot t}$$

where $f_{g,t,i}$ is the frequency of minority variants present in nucleotide site $i$ for gene segment $g$ and $n_{g,t}$ is the number of all available sites (**Xue and Bloom, 2020**). Distinct rates were calculated for synonymous and nonsynonymous iSNVs. If a variant was found in overlapping reading frames and a nonsynonymous change was observed in any of those frames, it would be accounted for as a nonsynonymous mutation. The corresponding whole-genome evolutionary rate ($r_t$) on day $t$ is computed by summing the rates across all gene segments:

$$r_t = \sum_g r_{g,t}$$

## Haplotype reconstruction

The most parsimonious viral haplotypes of each gene segment were reconstructed by fitting the observed nucleotide variant count data to a Dirichlet multinomial model using a previously developed ML approach to infer haplotype frequencies (**Ghafari et al., 2020**). Assuming that the viral population is made up of a set of $K$ haplotypes with frequencies $q_k$, the observed partial haplotype frequencies $q_l$ for a polymorphic site $l$ can be computed by multiplying a projection matrix $T_l$. For instance, if the set of hypothetical full haplotypes is assumed to be {$AA$, $GA$, $AG$}, the observed partial haplotype frequencies for site $l = 1$, $q_{A-}$ and $q_{G-}$ are computed as

$$q_l = T_l q_k \Rightarrow \begin{bmatrix} q_{A-} \\ q_{G-} \end{bmatrix} = \begin{bmatrix} ccc1 & 0 & 1 \\ ccc0 & 1 & 0 \end{bmatrix} \times \begin{bmatrix} q_{AA} \\ q_{GA} \\ q_{AG} \end{bmatrix}$$

A list of potential full haplotypes was generated from all combinations of nucleotide variants observed in all polymorphic sites of the gene segment. Starting from $K = 1$ full haplotype, the optimal full haplotype frequency $q_k$ is inferred by maximizing the likelihood function:

$$LL = \sum_l \log \mathcal{L}(x_l | T_l q_k, \varphi)$$

where $\mathcal{L}$ is Dirichlet multinomial likelihood, $x_l$ is the observed variant count data for read type $l$, and $\varphi$ is the overdispersion parameter, assumed to be $1 \times 10^{-3}$. Simulated annealing was used to optimize the haplotype frequencies by running two independent searches for at least 5000 states (iterations) until convergence was reached. In each state, the distribution of $q_k$ was drawn from a Gaussian distribution centered at the frequency distribution of the previous state with a standard deviation of 0.05. One additional haplotype was added to the set of $K$ full haplotypes during each round of optimization.

The resulting $K$ haplotypes reconstructed depend on the order in which the list of potential full haplotypes is considered. As mentioned above, paired-end reads were merged to produce longer reads (up to ~500–600 base pairs) for mapping in the case of the seasonal A/H3N2 virus samples. Additionally, the single-stranded A/H1N1pdm09 viral reads generated from 454 sequencing can be as long as ~500 base pairs. Consequently, there was a non-trivial number of reads where co-mutations were observed in multiple polymorphic sites. Since iSNV frequencies are generally low, haplotypes with co-mutating sites would inevitably be relegated to the end of the list order if ranked by their expected joint probabilities. As such, the list of full potential haplotypes was ordered in descending order based on the score of each full haplotype set $k$ ($s_k$):

$$s_k = f_{ss,k} \times f_{ms,k}$$

where $f_{ss,k}$ and $f_{ms,k}$ are both joint probabilities of the full haplotype $k$ computed in different ways. $f_{ss,k}$ is the expected joint probability frequency calculated from the observed independent frequencies of each variant for each polymorphic site found in the full haplotype $k$. $f_{ms,k}$ is based on the observed frequencies of variants spanning across the sets of highest hierarchal combination of polymorphic sites ($f_{ms,k}$).

For example, given a segment where iSNVs were found in three sites, the following reads were mapped: (A, A, C), (T, A, C), (A, T, C), (A, C, –), (–, A, C), and (–, T, C). We can immediately see that the top hierarchal combination of polymorphic sites (i.e., possible haplotypes) are (A, A, C), (T, A, C), and (A, T, C) (i.e., we would compute $f_{ms,(A,A,C)}$, $f_{ms,(T,A,C)}$, and $f_{ms,(A,T,C)}$, respectively). The observed number of reads with (–, A, C) will be counted towards the computation of both $f_{ms,(A,A,C)}$ and

$f_{ms,(T,A,C)}$ since they could be attributed to either haplotype. Similarly, reads with (–, T, C) will be absorbed towards the counts to compute $f_{ms,(A,T,C)}$. Finally, we see that reads with (A, C, –) are not a subset of any of the top hierarchal haplotypes considered. As such, they form the fourth possible top hierarchal haplotype on its own. As such, if we were to compute the ranking for haplotype (A, A, C):

$$\begin{aligned} s_{(A,A,C)} &= f_{ss,(A,A,C)} \times f_{ms,(A,A,C)} \\ &= \left\{ f_{(A,-,-)} \times f_{(-,A,-)} \times f_{(-,-,C)} \right\} \times f_{ms,(A,A,C)} \end{aligned}$$

If any nucleotide variants in the observed partial haplotypes were unaccounted for in the current round of full haplotypes considered, they were assumed to be generated from a cloud of 'noise' haplotypes that were present in no more than 1%. Bayesian information criterion (BIC) was computed for each set of full haplotypes considered, and the most parsimonious set of $K$ haplotypes was determined by the lowest BIC value.

## Linkage disequilibrium

Using the estimated frequencies of the most parsimonious reconstructed haplotypes, conventional Lewontin's metrics of linkage disequilibrium were computed to detect for potential epistatic pairs of nonsynonymous variants:

$$LD_{ij} = \hat{q}_{ij} - \hat{q}_i \hat{q}_j$$

where $\hat{q}_i$ and $\hat{q}_j$ are the estimated site-independent iSNV frequencies of sites $i$ and $j$, respectively, while $\hat{q}_{ij}$ is the frequency estimate of variants encoding co-variants in both $i$ and $j$. Dividing $LD$ by its theoretical maximum normalizes the linkage disequilibrium measure:

$$LD' = \frac{LD}{LD_{max}}$$

$$LD_{max} = \begin{cases} \max\left\{ -\hat{q}_i \hat{q}_j, -(1-\hat{q}_i)(1-\hat{q}_j) \right\} & if\, LD > 0 \\ \min\left\{ \hat{q}_i (1-\hat{q}_j), (1-\hat{q}_i) \hat{q}_j \right\} & if\, LD < 0 \end{cases}$$

## FoldX analyses

FoldX (https://foldxsuite.crg.eu/) was used to estimate structural stability effects of likely linked nonsynonymous minority variants found in the NP of within-host A/H3N2 virus populations. At the time of writing of this paper, there was no A/H3N2-NP structure available. Although the eventual NP structure (PDB: 3ZDP) adopted for stability analyses was originally derived from H1N1 virus (A/WSN/33) (*Chenavas et al., 2013*), it was the most well-resolved (2.69 Å) crystal structure available, with 78.5% amino acid identity relative to the NP of A/Brisbane/10/2007. Previous work has shown that the mutational effects predicted by FoldX using an NP structure belonging to A/WSN/33 (H1N1) were similar to those experimentally determined on a A/Brisbane/10/2007 NP (*Ashenberg et al., 2013*). FoldX first removed any potential steric clashes to repair the NP structure. It then estimated differences in free energy changes as a result of the input amino acid mutation (i.e., $\Delta\Delta G = \Delta G_{mutant} - \Delta G_{wild-type}$) under default settings (298 K, 0.05 M ionic strength, and pH 7.0). Five distinct simulations were made to estimate the mean and standard deviation $\Delta\Delta G$ values.

## Within-host simulations

We implemented forward-time Monte Carlo simulations with varying population size using a simplified within-host evolution model to test if our hypotheses could explain the different evolutionary dynamics observed between A/H3N2 and A/H1N1 viral populations. We assumed that a single virion leads to a productive influenza virus infection within an individual and computed changes in the virus population size ($N$) using a target cell-limited model. New virions are produced upon infection by existing virions at a rate of $\beta CN$, where $C$ is the existing number of target cells while $\beta$ is the rate of per-cell per-virion infectious contact. Upon infection, a cell will produce $r$ number of virions before it is rendered unproductive. We assume that infected individuals did not mount any antibody-mediated immune response, setting the virus' natural per capita decay rate ($d$) such that virions continue

to be present within host for 14 days (*Figure 6—figure supplement 1* and *Table 2*). $\beta$ is then computed by fixing the within-host basic reproduction number ($R_0$):

$$R_0 = \frac{\beta C_0 r}{d}$$

where $C_0$ is the initial (maximum) target cell population size. We solve the following system of ordinary differential equations numerically to compute the number of virions per viral replicative generation ($N(t)$):

$$\frac{dC}{dt} = -\beta CN$$

$$\frac{dN}{dt} = \beta CN - dN$$

We assume a binary genome of length $L$, distinguishing between synonymous and nonsynonymous loci. For A/H3N2 viruses, we hypothesized that most de novo mutations are either weakly deleterious or neutral. To estimate the number of such sites, we aligned A/H3N2 virus sequences that were collected between 2007 and 2012 and identified all polymorphic sites with variants that did not fixate over time (i.e., <95% frequency over 1-month intervals). We estimated $L = 1050$ with 838 and 212 synonymous and nonsynonymous loci, respectively.

We tracked the frequency distribution of genotypes present for every generation $t$. We assumed that mutations occur at per-locus, per-generation rate μ. During each generation $t$, the number of virions incurring a single-locus mutation followed a Poisson distribution with mean $N(t)\mu L$. For each virion, the mutant locus was randomly selected across all loci. We assumed that all synonymous and a fraction of nonsynonymous sites ($f_{neu,NS}$) are neutral (i.e., (log) fitness effect $s = 0$). The remaining nonsynonymous sites either had an additive deleterious ($s_d$) or beneficial ($s_b$) fitness effect when mutated. The magnitude of $s_d$ / $s_b$ follow an exponential distribution with mean effect $|\bar{s}|$. Epistasis was neglected throughout. The distribution of genotypes in the next generation $t + 1$ was achieved by resampling individuals according to Poisson distribution with mean $N(t + 1)P_f(g, t)$, where $P_f(g, t)$ is the relative fitness distribution of genotype $g$ during generation $t$.

To decrease the computational costs of the simulations, specifically when $N(t)$ reaches orders of $10^{10} - 10^{11}$ virions (*Figure 6—figure supplement 1*), we implemented an upper population size limit of $10^7$ virions. Given the mutation rate assumed (*Table 2*), $N(t)\mu \gg 1$ for $N(t) \geq 10^7$ virions, mutation-selection balance is theoretically expected for a single-locus (deleterious) mutant model (Appendix A5). We ran 500 simulations for each variable set of $f_{neu,NS}$ and $s_d$ / $s_b$ values. All parameter values used in the model are given in *Table 2*.

## Phylogenetic inference

All ML phylogenetic trees were reconstructed with IQTREE (v. 1.6.10) (*Nguyen et al., 2015*) using the GTR+I+G4 nucleotide substitution model.

**Table 2.** Parameter values used in the within-host model.

| Parameter | Meaning | Value (units) | Source |
|---|---|---|---|
| - | Number of hours per replicative generation | 6 hr | Assumption |
| $r$ | Average number of virions produced by an infected cell | 100 virions | *Frensing et al., 2016* |
| $C_0$ | Initial target cell population size | $4 \times 10^8$ virions | *Hadjichrysanthou et al., 2016* |
| $d$ | Per capita decay rate | Two per-generation | Assumption |
| $R_0$ | Within-host basic reproduction number | 5 | *Hadjichrysanthou et al., 2016* |
| μ | Per-site, per-generation mutation rate | $3 \times 10^{-5}$ per-site, per-generation | *McCrone et al., 2020* |

## Data availability

All raw sequence data have been deposited at NCBI sequence read archive under BioProject Accession number PRJNA722099. All custom Python code and Jupyter notebooks to reproduce the analyses in this paper are available online: https://github.com/AMC-LAEB/Within_Host_H3vH1 (copy archived at swh:1:rev:44e44ddbfab4d157a3c5efd559972f51dec6454a), *Han, 2021*.

## Acknowledgements

We thank Carolien van de Sandt for helpful discussions. We gratefully acknowledge the authors and originating and submitting laboratories (*Supplementary file 5*) for the reference sequences retrieved from GISAID's EpiFlu Database used in this study. AXH, ZCFG, and CAR were supported by ERC NaviFlu (no. 818353). The South East Asia Infectious Disease Clinical Research Network (SEAICRN) was funded by the National Institutes of Allergy and Infectious Diseases, National Institutes of Health (US), N01-A0-50042, HHSN272200500042C.

## Additional information

### Funding

| Funder | Grant reference number | Author |
|---|---|---|
| H2020 European Research Council | 818353 | Alvin X Han<br>Zandra C Felix Garza<br>Colin A Russell |
| National Institute of Allergy and Infectious Diseases | N01-A0-50042 | Matthijs RA Welkers<br>René M Vigeveno<br>Nhu Duong Tran<br>Thi Quynh Mai Le<br>Thai Pham Quang<br>Dinh Thoang Dang<br>Thi Ngoc Anh Tran<br>Manh Tuan Ha<br>Thanh Hung Nguyen<br>Quoc Thinh Le<br>Thanh Hai Le<br>Thi Bich Ngoc Hoang<br>Kulkanya Chokephaibulkit<br>Pilaipan Puthavathana<br>Van Vinh Chau Nguyen<br>My Ngoc Nghiem<br>Van Kinh Nguyen<br>Tuyet Trinh Dao<br>Tinh Hien Tran<br>Heiman FL Wertheim<br>Peter W Horby<br>Annette Fox<br>H Rogier van Doorn<br>Dirk Eggink<br>Menno D de Jong |
| National Institutes of Health | HHSN272200500042C | Matthijs RA Welkers<br>René M Vigeveno<br>Nhu Duong Tran<br>Thi Quynh Mai Le<br>Thai Pham Quang<br>Dinh Thoang Dang<br>Thi Ngoc Anh Tran<br>Manh Tuan Ha<br>Thanh Hung Nguyen<br>Quoc Thinh Le<br>Thanh Hai Le<br>Thi Bich Ngoc Hoang<br>Kulkanya Chokephaibulkit<br>Pilaipan Puthavathana<br>Van Vinh Chau Nguyen<br>My Ngoc Nghiem<br>Van Kinh Nguyen |

Tuyet Trinh Dao
Tinh Hien Tran
Heiman FL Wertheim
Peter W Horby
Annette Fox
H Rogier van Doorn
Dirk Eggink
Menno D de Jong

The funders had no role in study design, data collection and interpretation, or the decision to submit the work for publication.

## Author contributions

Alvin X Han, Conceptualization, Data curation, Software, Formal analysis, Validation, Investigation, Visualization, Methodology, Writing - original draft, Writing - review and editing; Zandra C Felix Garza, Conceptualization, Data curation, Formal analysis, Validation, Investigation, Visualization, Methodology, Writing - original draft, Writing - review and editing; Matthijs RA Welkers, Conceptualization, Resources, Data curation, Software, Formal analysis, Validation, Investigation, Methodology, Writing - review and editing; René M Vigeveno, Resources, Data curation, Validation, Investigation, Methodology, Writing - review and editing; Nhu Duong Tran, Thi Quynh Mai Le, Thai Pham Quang, Dinh Thoang Dang, Thi Ngoc Anh Tran, Manh Tuan Ha, Thanh Hung Nguyen, Quoc Thinh Le, Thi Bich Ngoc Hoang, Kulkanya Chokephaibulkit, Pilaipan Puthavathana, Van Vinh Chau Nguyen, My Ngoc Nghiem, Van Kinh Nguyen, Tuyet Trinh Dao, Tinh Hien Tran, Heiman FL Wertheim, Peter W Horby, Annette Fox, H Rogier van Doorn, Resources, Data curation, Funding acquisition, Investigation, Project administration, Writing - review and editing; Thanh Hai Le, Resources, Data curation, Funding acquisition, Investigation, Project administration; Dirk Eggink, Conceptualization, Data curation, Formal analysis, Supervision, Investigation, Visualization, Methodology, Writing - review and editing; Menno D de Jong, Conceptualization, Resources, Data curation, Supervision, Funding acquisition, Investigation, Methodology, Project administration, Writing - review and editing; Colin A Russell, Conceptualization, Formal analysis, Supervision, Funding acquisition, Validation, Investigation, Visualization, Methodology, Writing - original draft, Project administration, Writing - review and editing

## Author ORCIDs

Alvin X Han ![iD] https://orcid.org/0000-0001-6281-8498
Zandra C Felix Garza ![iD] http://orcid.org/0000-0001-7262-2165
Matthijs RA Welkers ![iD] https://orcid.org/0000-0002-2982-0278
Thai Pham Quang ![iD] http://orcid.org/0000-0002-3796-6162
Annette Fox ![iD] http://orcid.org/0000-0002-0565-7146

## Ethics

Human subjects: The Institutional Review Board of all hospitals, the National Institute of Allergy and Infectious Diseases, and the Oxford Tropical Research Ethics Committee approved the study. Written informed consent was given by all patients (or proxies).

## Decision letter and Author response

Decision letter https://doi.org/10.7554/eLife.68917.sa1
Author response https://doi.org/10.7554/eLife.68917.sa2

## Additional files

### Supplementary files

• Supplementary file 1. Mean number of nonsynonymous (NS), synonymous (S), and stop codon (Stop) variants per sample for each gene segment as well as the corresponding NS/S ratio.

• Supplementary file 2. Potentially linked nonsynonymous variants in within-host A/H1N1pdm09 and A/H3N2 virus samples. Sample names are given in the format of 'Patient ID_Days since symptom

onset.' Both linkage disequilibrium ($LD$) and the normalized $LD'$ measures are tabulated alongside the inferred maximum-likelihood haplotype frequencies ($q_{10}$ and $q_{01}$ are the haplotype frequencies with variant i or ii only while $q_{11}$ is the frequency of haplotypes encoding both variants).

- Supplementary file 3. A/H3N2 segment-specific primers.
- Supplementary file 4. Patients metadata (provided as an Excel file).
- Supplementary file 5. Acknowledgment table of reference sequences downloaded from GISAID.
- Transparent reporting form

### Data availability

All raw sequence data have been deposited at NCBI sequence read archive under BioProject Accession number PRJNA722099. All custom Python code and Jupyter notebooks to reproduce the analyses in this paper are available online: https://github.com/AMC-LAEB/Within_Host_H3vH1 (copy archived at https://archive.softwareheritage.org/swh:1:rev:44e44ddbfab4d157a3c5efd559972f51dec6454a).

The following dataset was generated:

| Author(s) | Year | Dataset title | Dataset URL | Database and Identifier |
|---|---|---|---|---|
| South East Asia Infectious Disease Clinical Research Network | 2021 | Deep sequencing of virological samples from patients infected with influenza A viruses | https://www.ncbi.nlm.nih.gov/bioproject/PRJNA722099 | NCBI BioProject, PRJNA722099 |

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

## Appendix 1

### A1: hemagglutinin and neuraminidase minority variants in A/H3N2 virus samples

The amino acid variants emerging in the HA and NA proteins of A/H3N2 virus samples were discussed in a previous work (*Koel et al., 2020*). Briefly, given the lack of antibody-mediated immune response in this cohort of mostly naïve children, HA amino acid variants emerging in putative antigenic sites were generally low in frequencies (median frequency = 0.04, IQR = 0.03–0.06) and all only became detectable 3–4 days post-symptom onset (*Figure 4—figure supplement 1*). Notably, two of these intra-host mutations were also found in the global A/H3N2 virus population in high frequencies: HA-S45N and -D53N (H3 numbering without signal peptide). Both mutations are part of the canonical antigenic site C of H3 and emerged within separate individuals, with D53N eventually becoming the majority variant (97%) as late as day 13 post-symptom onset in one of them. Oseltamivir-resistant amino acid mutations E119V, R292K, and N329K arose in 10 patients that were treated with the antiviral drug and mostly rose in within-host frequencies only 4–7 days after administration of oseltamivir (*Figure 4—figure supplement 2*).

### A2: genetic diversity by $\pi$ statistic

Given that the number of identified iSNVs can potentially be biased by variations in sequencing coverage, genetic diversity was also assessed using nucleotide diversity $\pi$ statistic (*Nei and Li, 1979*). This approach constitutes a more robust measure of within-host diversity as it is solely dependent on the underlying variant frequencies. While the number of polymorphic sites provides an estimate of 'richness' in the viral population, it may be incompatible to compare iSNV counts between samples with different read coverage profiles (*Zhao and Illingworth, 2019*). In contrast, $\pi$ is a more robust metric that is unbiased by sequencing depth. For each site $l$:

$$\pi_l = \frac{N_l(N_l - 1) - \sum_j n_{j,l}(n_{j,l} - 1)}{N_l(N_l - 1)}$$

where $N_l$ and $n_{j,l}$ are the coverage and number of reads encoding allele $j$ in site $l$, respectively (*Zhao and Illingworth, 2019*). To compute the $\pi$ statistic for the entire genome of length $L$:

$$\pi = \sum_{l=1}^{L} \frac{\pi_l}{L}$$

Here, we observed similar trends in genetic diversity when using $\pi$ statistics compared to iSNVs counts (*Appendix 1—figure 1*). $\pi$ weakly increased with respect to time and Ct values for A/H3N2 viruses (days since illness onset: Spearman $\rho = 0.388$, $p = 1.52 \times 10^{-8}$; CT: $\rho = 0.455$, $p = 3.66 \times 10^{-10}$) while remaining relatively invariant for A/H1N1pdm09 viruses (days since symptom onset: $\rho = 0.017$, $p = 0.92$; CT: $\rho = 0.240$, $p = 0.13$).

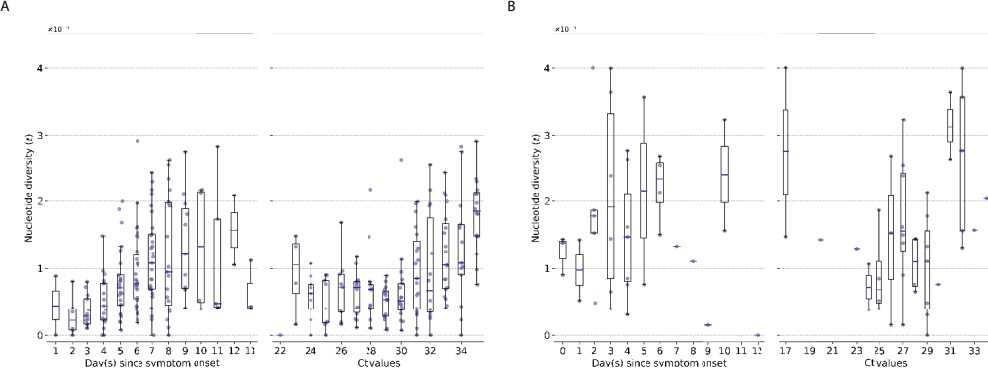

**Appendix 1—figure 1.** Genetic diversity of within-host influenza A virus populations as estimated by
*Appendix 1—figure 1 continued on next page*

*Appendix 1—figure 1 continued*

nucleotide diversity $\pi$ statistic. Box plots summarizing the $\pi$ statistic (intra-host single-nucleotide variants [iSNVs]; median, interquartile range [IQR], and whiskers extending within median $\pm 1.5$ $\times$ IQR) computed for samples with adequate breadth of coverage across the whole influenza genome. (**A**) Seasonal A/H3N2 and (**B**) pandemic A/H1N1pdm09 viruses. All box plots are either stratified by day(s) since symptom onset or qPCR cycle threshold (Ct) values.

## A3: potential linked minority variants in within-host virus populations

For both within-host seasonal A/H3N2 and pandemic A/H1N1 virus populations, there were few instances of potentially linked nonsynonymous variants, and if such co-variants were to exist, they were mainly found in the internal gene segments (*Supplementary file 2*). There was only one pair of HA amino acid mutations (E261G/L455F) that was encoded by a minority haplotype of A/ H1N1pdm09 viruses infecting one individual, but the normalized Lewontin's linkage disequilibrium measure ($LD'$) was less than 0.5, suggesting a low likelihood that the mutation pair was linked non-randomly. These potentially linked variants tend to emerge late in the infection (6–7 days post-illness onset) for both viral subtypes, in inferred haplotypes appearing at low frequencies within-host (median frequency = 0.08, IQR = 0.03–0.12) that were not shared between multiple individuals.

## A4: transmission bottleneck size estimation of pandemic A/ H1N1pdm09 viral infections

Index cases were previously identified for six of the seven households where the pandemic A/ H1N1pdm09 viral samples were collected (*Thai et al., 2014*). Assuming that the non-index cases within the same household were secondarily infected by the index case, five transmission pairs were identified where samples with adequate breadth of coverage (>70% of genome covered with >50× coverage; *Appendix 1—figure 2*) were collected from the index patient on an earlier date relative to the secondary case.

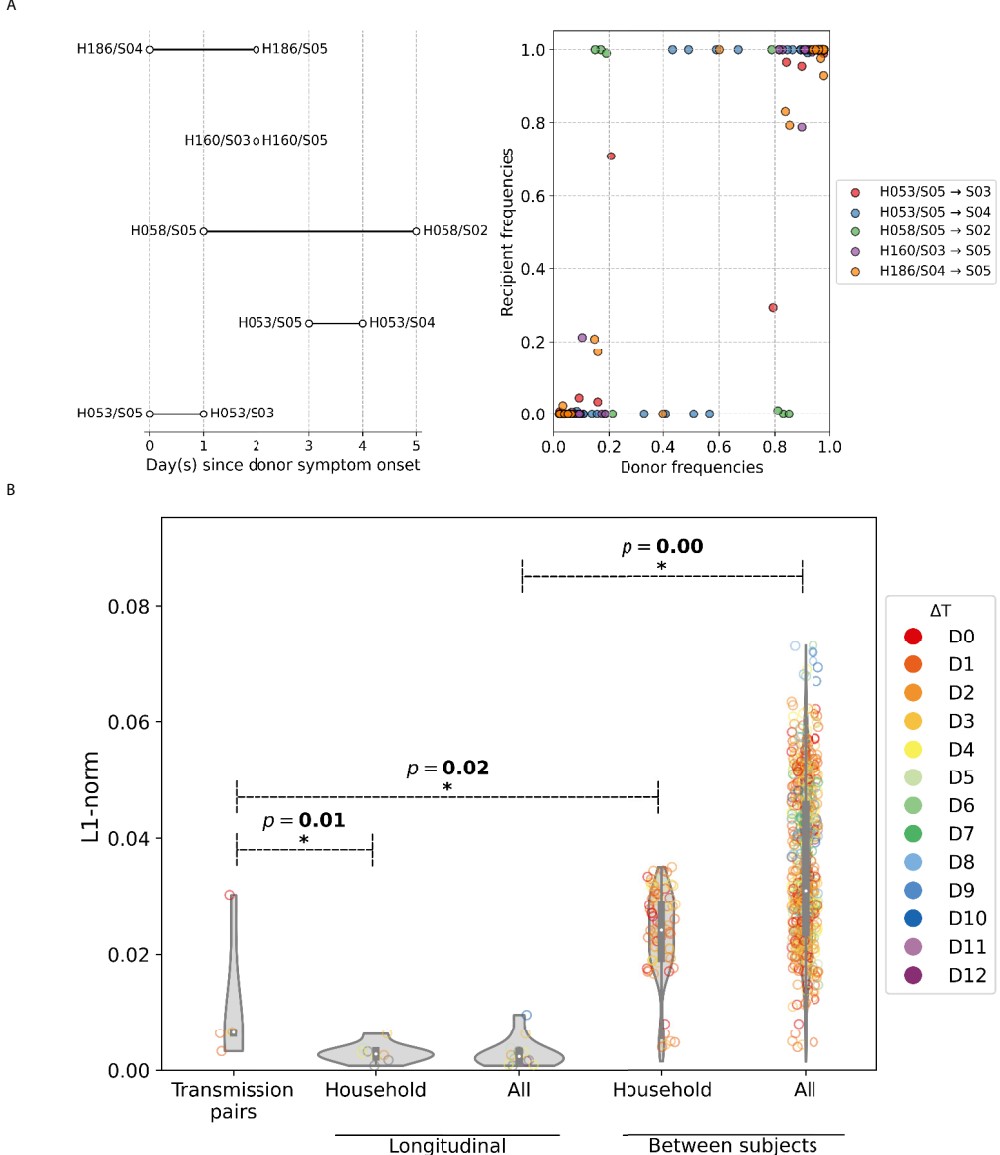

**Appendix 1—figure 2.** A/H1N1pdm09 virus household transmission pairs. (**A**) Schematic of A/H1N1pdm09 virus household transmission pairs identified by epidemiological linkage and plotted based on timing of sample collection (left panel) Intra-host single-nucleotide variant (iSNV) frequencies found in the donors and recipients of the five transmission pairs (right panel). (**B**) Violin plots of L1-norm pairwise genetic distance per site between different A/H1N1pdm09 virus sample pairs (each circle = 1 pair of virus samples). Transmission pairs are those represented in *Figure 5A*. Longitudinal pairs are made up of sample pairs collected from the same individual on the first and any other later timepoints on which the patient was sampled. These pairs are stratified by whether they were collected from households located in the same community (i.e., household) or combined with the rest of the analyzed A/H1N1pdm09 virus samples collected from hospitals (i.e., all). For the same aforementioned categories, we also plotted the distribution of L1-norm distances for pairs of viruses collected from different individuals. All circles are colored by the difference in days between which the sample pairs were collected ($\Delta T$). All p-values reported are based on Mann–Whitney U tests, which were used to determine if the L1-norm genetic distance distributions of the two categories marked by the ends of the horizontal line above are statistically distinct.

Virus transmission bottleneck sizes were then estimated using two binomial sampling models that were elaborated in detail by *Sobel Leonard et al., 2017* and *McCrone et al., 2018*. First, the

presence/absence model computes transmission probability as the probability that a transmitted donor iSNV was found in at least one genome in the bottleneck population:

$$P_{d,i}(A|N_b) = p_{d,A}^{N_b}$$

where $A$ refers to the transmitted iSNV in polymorphic site $i$, $N_b$ is the bottleneck size, and $p_{d,A}$ is the frequency of allele $A$ in the sampled virus population within donor $d$.

The presence/absence model does not incorporate recipient frequencies of transmitted iSNVs in its probability calculations. It assumes that all transmitted iSNVs are detected in the recipient, and thus any donor iSNVs that are not present in the recipient are considered to have not been transmitted. It also does not account for any changes to $p_d$ between the time of sampling and day of transmission. The ML estimate of $N_b$ would thus yield the largest log likelihood value given by

$$LL(N_b) = \sum_d \sum_i \ln P_{d,i}$$

To incorporate information on recipient frequencies that can change between transmission and sampling, transmission bottleneck sizes were re-estimated using a second beta-binomial model formulated by *Sobel Leonard et al., 2017*. For each allele $A$ observed in polymorphic site $i$ that was transmitted from donor $d$ to recipient $r$, the log-likelihood of $N_b$ is given as

$$LL(N_b)_{d,r}^{transmitted} = \sum_{A_i} \ln \left\{ \sum_{k=1}^{N_b} p_{beta}\left(p_{r,A_i}|k, N_b - k\right) p_{bin}\left(k|N_b, p_{d,A_i}\right) \right\}$$

where $p_{beta}\left(p_{r,A_i}|k, N_b - k\right)$ is the conditional probability density, as modelled by the beta distribution, that the transmitted iSNV, $A_i$, is found in the recipient at frequency $p_{r,A_i}$ given that the variant is found present in $k$ genomes out of the total transmission bottleneck of $N_b$ genomes. $p_{bin}\left(k|N_b, p_{d,A_i}\right)$ is the binomial probability of drawing $k$ genomes with allele $A_i$ in a sample of $N_b$ genomes and variant frequency of $p_{d,A_i}$ within the donor.

As some of the iSNVs in the donor may not be transmitted or were present below the 2% minimum variant frequency cutoff, the likelihood of these events was computed by

$$LL(N_b)_{d,r}^{lost} = \sum_{A_i} \ln \left\{ \sum_{k=1}^{N_b} p_{beta,cdf}\left(p_{r,A_i} < 0.02 | k, N_b - k\right) p_{bin}\left(k|N_b, p_{d,A_i}\right) \right\}$$

where $p_{beta,cdf}$ is the cumulative distribution function of the beta distribution.

The ML estimate of $N_b$ as described by the beta-binomial was then computed by searching for the value of $N_b$ that would give the largest value of

$$LL(N_b) = \sum_{d,r} LL(N_b)_{d,r}^{transmitted} + LL(N_b)_{d,r}^{lost}$$

Log-likelihood values were computed for a range $N_b$ between 1 and 1000 genomes for both models.

Transmission bottleneck size was then estimated by aggregating over all transmission pairs and applying two previously developed sampling-based models (*McCrone et al., 2018*; *Sobel Leonard et al., 2017*). We estimated the transmission bottleneck of pandemic A/H1N1pdm09 viruses to be 1–2 genomes (ML presence-absence model estimate = 1 genome; ML beta-binomial model estimate = 2 genomes). The tight bottleneck was further reflected in the iSNV frequency plot where most of the donor variants were either present as the single majority allele or were not transmitted/undetected in the recipient, with few shared iSNVs between the two virus populations (*Appendix 1—figure 2A*).

Furthermore, we observed that the virus populations between patients (median L1-norm distance = 0.031 divergence per site, IQR = 0.024–0.046 divergence per site) were significantly more different than longitudinal samples collected from the same individual (median L1-norm distance = $2.35 \times 10^{-3}$ divergence per site, IQR = $1.54 \times 10^{-3} - 3.18 \times 10^{-3}$ divergence per site), suggesting that the haplotypes found within an individual remain relatively invariant throughout the course of infection (*Appendix 1—figure 2B*). Combined with the fact that the per-site L1-norm genetic distance

between samples attributed to the identified transmission pairs (median L1-norm distance = $6.49 \times 10^{-3}$ divergence per site, IQR = $6.34 \times 10^{-3} - 6.63 \times 10^{-3}$ divergence per site) was greater than those computed for longitudinal sample pairs of each household individual (median L1-norm distance = $2.77 \times 10^{-3}$ divergence per site, IQR = $2.18 \times 10^{-3} - 3.32 \times 10^{-3}$ divergence per site; Mann–Whitney U p-value=0.01; *Appendix 1—figure 2B*), it is likely that only a limited number of haplotypes were shared between individuals in a transmission pair. Based on the most parsimonious reconstructed haplotypes for the five transmission pairs encoding shared iSNVs, we estimated the median number of haplotypes transmitted from donor to recipient to be between 1 and 2 haplotypes.

## A5: mutation-selection balance

Considering a single-locus mutant with deleterious fitness effect $s$ (i.e., $s<0$), the frequency of the mutant allele ($f$) can be modeled by the following stochastic differential equation, otherwise known as the Langevin equation (*Good and Desai, 2013*):

$$\underbrace{\frac{\partial f}{\partial t} = sf(1-f)}_{\text{selection}} + \underbrace{\sqrt{\frac{f-(1-f)}{N}}\eta(t)}_{\text{genetic drift}} + \underbrace{\mu(1-f)}_{\text{mutation}}$$

where $N$ is the population size, $\mu$ is the mutation rate, and $\eta(t)$ is the stochastic noise term due to genetic drift. For any Langevin equation, we can find the time-dependent probability distribution of $f$ (i.e., $\frac{\partial p(f,t)}{\partial t}$) by its corresponding Fokker–Planck equation. At stationarity (i.e., $\frac{\partial p(f,t)}{\partial t} = 0$), its solution is known to be

$$p(f) \propto \frac{e^{-2N\Lambda(f)}}{f(1-f)}$$

where $-\frac{\partial \Lambda(f)}{\partial t} = s + \frac{\mu}{f}$

Integrating $-\frac{\partial \Lambda(f)}{\partial t}$, we will get

$$-\Lambda(f) = sf + \mu \ln f$$

which can then be substituted to obtain

$$p(f) \propto e^{2Nsf} \cdot f^{2N\mu - 1}$$

In other words, $p(f)$ strongly depends on $N\mu$. If $N\mu \gg 1$, we can see that $p(f)$ will strongly peak at some characteristic value of $f$ that minimizes $\Lambda(f)$. If $N$ is large, we can assume that drift effects are negligible and $f$ is largely deterministic due to selection:

$$\frac{\partial f}{\partial t} = sf(1-f) + \mu(1-f) = \left\{ -\frac{\partial \Lambda(f)}{\partial t} \right\} [f(1-f)]$$

$$\Rightarrow \frac{\partial \Lambda}{\partial t} = \frac{\partial f}{\partial t}\left(\frac{\partial \Lambda}{\partial f}\right) = \frac{\partial f}{\partial t}\left\{ -\frac{\partial f}{\partial t}\left(\frac{1}{f(1-f)}\right) \right\} \leq 0$$

In other words, selection dynamics minimizes the $\Lambda(f)$ term, and as such, result in mutation-selection balance:

$$-\frac{\partial \Lambda(f)}{\partial t} = s + \frac{\mu}{f} = 0$$

$$\Rightarrow f = -\frac{\mu}{s}$$

