## [Decision Letter]

**Acceptance summary:**

This paper provides important observations and insights on the within-host evolution of seasonal influenza, particularly H3N2 in children. As in adults, H3N2 appears to evolve largely by purifying selection during the initial stages of infection when transmission is likely to occur, and nonsynonymous mutations accumulate later. The authors propose a model of mutation-selection balance that might explain diversity in young children with longer infections.

**Decision letter after peer review:**

Thank you for submitting your article "Within-host evolutionary dynamics of seasonal and pandemic human influenza A viruses in young children" for consideration by *eLife*. Your article has been reviewed by 3 peer reviewers, one of whom is a member of our Board of Reviewing Editors, and the evaluation has been overseen by Miles Davenport as the Senior Editor. The reviewers have opted to remain anonymous.

Essential revisions:

This study adds to our knowledge of the evolutionary dynamics of influenza within hosts. The longitudinal data – from likely primary H3N2 infections in children and infections with emerging H1N1pdm09 – are interesting, and the evolutionary analysis has laudably broad scope. That said, the reviewers agreed that limitations in both the data and analysis cast doubt on the conclusions. More analysis could strengthen the study. After consultation, our major concerns are that:

1. Not enough attention is given to potential errors affecting the sequences. Reviewers discussed both systematic sequencing and random PCR errors. Ideally plasmid controls would have been used, but under the circumstances, we suggest the authors instead perform sensitivity analysis and other checks recommended by Reviewers 2 and 3.

2. The temporal trends are not analyzed in a statistically careful way. Reviewers 1 and 3 raised concerns about the trends reported in Figure 2. (Are they really there?) This affects the general conclusions about H1N1pdm09 v. H3N2.

3. The simulations assume an odd distribution of fitness effects, which might skew the conclusions about the evolutionary regimes of the two subtypes. (It is not unclear why they should have contrasting evolutionary dynamics.) More thorough sensitivity analysis could help here.

The essential revisions would address these concerns. The reviewers were also in agreement on their more specific comments, which I hope you can use to strengthen the work.

*Reviewer #1 (Recommendations for the authors):*

My primary suggestion is effectively to clarify the statistics for the temporal trends. For instance, in Figure 2, do the p-values test for NS and S being significantly distinct from *each other* within a time point or gene or between time points/genes? I initially thought the latter given the claims in the text about temporal dynamics. I believe more tests are needed for these claims, not only for H1N1pdm09 (ll. 180-182). This will influence the comparison with simulation output. The Discussion might then also need updating (ll. 474-477).

*Reviewer #2 (Recommendations for the authors):*

Figure 1 figure supplement 2 appears to show variants present at reported frequencies that where not in all overlapping amplicons. These could be PCR artifacts or potentially real variants that were missed in one of the amplicons. Are the evolutionary rate dynamics driven by variants in this frequency range? Is there enough signal to filter out similar variants and validate the robustness of the findings?

Line 205: It is not clear how the frequency of synonymous mutations, by themselves indicates negative selection in the antigenic sites. What is the importance of the higher frequency of synonymous mutations found in antigenic sites?

Line 663: How were overlapping reading frames accounted for in the evolutionary rate calculation?

The qualitative similarity between the simulated and observed rates is nice addition to the manuscript. Is it possible to use the frequencies of mutations in longitudinal pairs to further support the hypothesis of mutational-selection balance?

*Reviewer #3 (Recommendations for the authors):*

1. The authors identify and analyze several recurrent mutations in the H3N2 M2 and NP genes, which were found in anywhere from 16 to 27 unlinked individuals. They argue that the recurrent NP mutation is a stabilizing mutation and is epistatically linked to several co-variants that may have destabilizing effects (Figure 5). I am concerned that these mutations may result from technical artifacts and may not represent genuine within-host variants, particularly since amino-acid variants that are known to be associated with oseltamivir resistance were only identified in two patients despite the administration of oseltamivir in many patients. In data from McCrone et al., 2018, for example, several sites appear to harbor low-frequency variants in unrelated individuals, much like the authors describe here, but those variants are also present in the plasmid controls, suggesting that they represent common, site-specific polymerase errors rather than recurrent mutations.

To try to determine whether these variants are technical errors, the authors would ideally sequence plasmid controls using the same protocols that were used to sequence the original samples. Since this is likely infeasible, the authors should also check their data to see if variation is present at M2-77, NP-G384R, and other apparent sites of recurrent mutation at frequencies below the variant-calling threshold. If variation is present at these sites in most samples at a frequency much higher than in neighboring sites, then this variation may reflect technical errors in sequencing. The authors called variants after mapping reads to a reference genome, but they might also try remapping to the sample consensus sequence to reduce the risk that mapping artifacts are causing these recurrent variant calls.

2. The authors write that non-synonymous mutations accumulate later in patient infections, but it's not clear to me how strongly the data in Figure 2A supports that argument, particularly given the limited number of samples collected at any given day following symptom onset. Does nonsynonymous diversity tend to increase over time in successive timepoints collected from the same individual? Does noise in variant frequencies account for the apparent fluctuations in synonymous diversity over time, and if so, how does noise affect the interpretation of nonsynonymous diversity?

3. In the simulations of A/H1N1pdm09 in Figure 6A and 6B, the authors simulate a distribution of fitness effects in which 1% of nonsynonymous mutations are neutral and the remaining mutations are weakly (s=0.01) or strongly (s=0.1) beneficial (lines 785-6). This distribution of fitness effects seems unrealistic to me – even for an emerging virus that may be adapting to a new host, most nonsynonymous mutations will still be deleterious because they affect basic protein functions. It's not clear to me that the large increase in nonsynonymous diversity that the authors observe from these simulations would be observed if deleterious mutations were adequately accounted for; the distribution would probably look much more similar to Figures 6D and 6E.

---

## [Author Response]

Essential revisions:This study adds to our knowledge of the evolutionary dynamics of influenza within hosts. The longitudinal data---from likely primary H3N2 infections in children and infections with emerging H1N1pdm09---are interesting, and the evolutionary analysis has laudably broad scope. That said, the reviewers agreed that limitations in both the data and analysis cast doubt on the conclusions. More analysis could strengthen the study. After consultation, our major concerns are that:1. Not enough attention is given to potential errors affecting the sequences. Reviewers discussed both systematic sequencing and random PCR errors. Ideally plasmid controls would have been used, but under the circumstances, we suggest the authors instead perform sensitivity analysis and other checks recommended by Reviewers 2 and 3.2. The temporal trends are not analyzed in a statistically careful way. Reviewers 1 and 3 raised concerns about the trends reported in Figure 2. (Are they really there?) This affects the general conclusions about H1N1pdm09 v. H3N2.3. The simulations assume an odd distribution of fitness effects, which might skew the conclusions about the evolutionary regimes of the two subtypes. (It is not unclear why they should have contrasting evolutionary dynamics.) More thorough sensitivity analysis could help here.The essential revisions would address these concerns. The reviewers were also in agreement on their more specific comments, which I hope you can use to strengthen the work.

We thank the editors for considering our manuscript for review. We have addressed all of the reviewers’ concerns and comments in detail below. The revised manuscript is submitted with tracked changes relative to the original submission.

Reviewer #1 (Recommendations for the authors):My primary suggestion is effectively to clarify the statistics for the temporal trends. For instance, in Figure 2, do the p-values test for NS and S being significantly distinct from each other within a time point or gene or between time points/genes? I initially thought the latter given the claims in the text about temporal dynamics. I believe more tests are needed for these claims, not only for H1N1pdm09 (ll. 180-182). This will influence the comparison with simulation output. The Discussion might then also need updating (ll. 474-477).

We thank the reviewer for their careful consideration of our manuscript.

We agree that there is a lack in statistical power in the A/H1N1pdm09 virus dataset to claim meaningful differences in temporal trends to A/H3N2 within-host dynamics. The only reasonable conclusion that can be made here is that there was a greater accumulation in nonsynonymous iSNVs relative to synonymous ones in A/H1N1pdm09 within-host virus populations. As per the reviewer’s suggestion, we have now removed the boxplots for the A/H1N1pdm09 virus panel in Figure 2B, replacing it with a scatter plot. We have also updated the manuscript to reflect our inability to characterise the within-host temporal trends for A/H1N1pdm09 viruses using this dataset:

Line 210: “We observed higher nonsynonymous evolutionary rates relative to synonymous ones initially after symptom onset but were unable to determine if they were significantly different due to the low number of samples (i.e. median = 2 samples per day post-symptom onset). In turn, we also could not meaningfully characterise the temporal trends of within-host evolution for the pandemic virus with this dataset. Nonetheless, consolidating over all samples across all time points, there was significantly higher rates of accumulation of nonsynonymous variants in the polymerase basic 2 (PB2), polymerase acidic (PA), HA and matrix (M) gene segments (Figure 2B, Figure 2 —figure supplement 2 and Figure 3 —figure supplement 2). All gene segments also yielded NS/S ratios > 1 (Table S1).”

Line 565: “Owing to the low number of A/H1N1pdm09 virus samples and different next-generation sequencing platforms used to sequence samples of the two virus subtypes and consequently differences in base calling error rates and depth of coverage (Figure 1 —figure supplement 1), we were unable to directly compare the observed levels of within-host genetic diversity and evolutionary dynamics between the two influenza subtypes here.”

Linderman et al., (PNAS, 2014) and Huang et al., (JCI, 2015) found that individuals born prior to the early 1980s possessed antibodies that recognized HA-166K (H3 numbering) residing in the Sa antigenic site of A/H1N1pdm09 viruses. They attributed this to previous exposures to seasonal A/H1N1 viruses with the HA-166K Sa epitope. This adaptive immune response likely led to the fixation of HA-K166Q in A/H1N1pdm09 viruses, which abrogated antibody recognition of this epitope. However, this epitope was shielded by glycans in seasonal A/H1N1 viruses in 1986 due to the acquisition of a glycosylation site in HA-129. As such individuals born after the late 1980s did not possess the same antibodies and are therefore unlikely to exert the same adaptive immune pressure as their older counterparts.

Out of the 32 A/H1N1pdm09-infected individuals analysed in our study, only six of them were born before 1986. The median birth year of all individuals was 1999 (IQR = 1989, 2005). Hence, the same adaptive immune pressure on HA-166K was not present in these younger individuals during the first wave of the A/H1N1pdm09 pandemic then. We also did not detect the HA-166Q variant in any of the six older individuals born prior to 1986.

Besides HA-166K, Li et al., (JEM, 2013) also found that individuals born between 1983 and 1996 have narrowly focused antibodies against the HA-133K epitope as a result of previous exposures to seasonal A/H1N1 viruses. HA-133K has, however, remained conserved in the global A/H1N1pdm09 virus population to date. We also did not find any variants above the calling threshold in any of the individuals investigated.

The HA protein is the primary target of human adaptive immune response, which in turn drives its antigenic evolution (Petrova and Russell, Nat Rev Microbiol, 2018). In terms of cellular immunity, HA encodes few CTL epitopes (Woolthuis et al., Sci Rep, 2016). Most CTL epitopes are found in the nucleoprotein (NP), which we have considered here in our discussion observing recurrent NP-G384R variants independently found in multiple individuals.

As mentioned earlier, the A/H1N1pdm09 virus dataset lack statistical power. As such, we are unable to characterise temporal trends for the pandemic virus and have no longer discuss this in the updated manuscript (see response to reviewer #3 as well).

However, the reviewer was right to point out one of our key conclusions that mutation-selection balance is only observed in naïve young children with longer A/H3N2 virus infections and would be less likely to hold for the typically shorter-lived infections of older children and adults. We have now put more emphasis on this conclusion in the abstract and discussion:

Line 42: “For A/H3N2 viruses in young children, early infection was dominated by purifying selection. As these infections progressed, nonsynonymous variants typically increased in frequency even when within-host virus titres decreased. Unlike the short-lived infections of adults where de novo within-host variants are rare, longer infections in young children allow for the maintenance of virus diversity via mutation-selection balance creating potentially important opportunities for within-host virus evolution.”

Line 530: “Through simulations of a within-host evolution model, we investigated the hypothesis that in the absence of any positive selection, the accumulation of nonsynonymous iSNVs was a result of their neutral or only weakly deleterious effects and the expanding within-host virion population size during later timepoints in longer infections of naïve young children such that mutation-selection balance was reached. In contrast, this balance was not detected in otherwise healthy older children or adults with short-lived influenza virus infections lasting no more than a week where de novo nonsynonymous iSNVs are rarely found ^4,8–11,44^.”

Reviewer #2 (Recommendations for the authors):Figure 1 figure supplement 2 appears to show variants present at reported frequencies that where not in all overlapping amplicons. These could be PCR artifacts or potentially real variants that were missed in one of the amplicons. Are the evolutionary rate dynamics driven by variants in this frequency range? Is there enough signal to filter out similar variants and validate the robustness of the findings?

As per the suggestion by the reviewer, we first excluded all iSNVs found under the 75^th^ percentile of frequency range of variants detected in only one of the overlapping. We then recomputed the day-by-day evolutionary rates (Figure 2 —figure supplement 3) and found similar relative rate dynamics as those presented in the main results (Figure 2A). We added the following text to manuscript:

Line 632: “We also performed additional checks to ensure that our results were not driven by potential PCR and/or technical artefacts. First, we excluded all iSNVs found under the 75^th^ percentile of frequency range of A/H3N2 variants that were found in only one of the overlapping amplicons. We then recomputed the daily within-host evolutionary rates with the remaining iSNVs (Figure 2 —figure supplement 3) and found that the relative temporal trends in synonymous and nonsynonymous rates remain similar to those in Figure 2A. We also checked that the distributions of frequencies for iSNVs found in recurrent mutation sites (i.e. NP-384 and M2-77) that are below variant calling threshold are comparable to those found in their neighbouring sites (±10 nucleotide positions; Figure 4 —figure supplement 2). Furthermore, we remapped the sample reads to their respective consensus sequences to minimize mapping of technical artefacts. We were still able to detect the recurring NP-G384R and M2-R77* amino acid mutations in multiple individuals and timepoints at similar frequencies when mapped to the reference genome (Figure 4 —figure supplement 1C-D and 3). As such, these recurrent mutations are unlikely to have been resulted from erroneous variant calls of artefacts.”

See response to reviewer #3 for further validation on the robustness of our variant calls.

Line 205: It is not clear how the frequency of synonymous mutations, by themselves indicates negative selection in the antigenic sites. What is the importance of the higher frequency of synonymous mutations found in antigenic sites?

Under neutral selection, the frequency of synonymous and nonsynonymous mutations should be similar to one another. However, if there was negative (purifying) selection to purge deleterious amino acid changes (nonsynonymous mutations) such as those that affect protein functions, this should result in an imbalance in mutation frequencies where there are relatively higher frequencies of synonymous mutations compared to nonsynonymous ones. As many antigenic sites of HA are close to the receptor-binding region, many nonsynonymous mutations in these sites could have negatives effects on host cell binding and thus be purged by negative selection.

Line 663: How were overlapping reading frames accounted for in the evolutionary rate calculation?

We considered a variant to be a nonsynonymous mutation if any of the overlapping reading frames engender nonsynonymous changes. This is now explicitly stated in the Methods (Line 783):

“If a variant was found in overlapping reading frames and a nonsynonymous change was observed in any of those frames, it would be accounted for as a nonsynonymous mutation.”

The qualitative similarity between the simulated and observed rates is nice addition to the manuscript. Is it possible to use the frequencies of mutations in longitudinal pairs to further support the hypothesis of mutational-selection balance?

Given that we are unable to quantify the range and level of uncertainty in variant frequencies of our dataset, it is difficult to use mutation frequencies to support the mutation-selection balance hypothesis. However, we observed that some nonsynonymous variants, including lethal stop-codon mutations such as M2-R77*, persisted in low frequencies within the same individual across multiple timepoints (Figure 4 —figure supplement 1). Furthermore, in response to reviewer #3, we performed linear regression on the computed evolutionary rates for each individual with samples collected at multiple timepoints to ascertain that the patterns observed were not because of aggregating the data from multiple individuals. See detailed response to reviewer #3 below.

Reviewer #3 (Recommendations for the authors):1. The authors identify and analyze several recurrent mutations in the H3N2 M2 and NP genes, which were found in anywhere from 16 to 27 unlinked individuals. They argue that the recurrent NP mutation is a stabilizing mutation and is epistatically linked to several co-variants that may have destabilizing effects (Figure 5). I am concerned that these mutations may result from technical artifacts and may not represent genuine within-host variants, particularly since amino-acid variants that are known to be associated with oseltamivir resistance were only identified in two patients despite the administration of oseltamivir in many patients. In data from McCrone et al., 2018, for example, several sites appear to harbor low-frequency variants in unrelated individuals, much like the authors describe here, but those variants are also present in the plasmid controls, suggesting that they represent common, site-specific polymerase errors rather than recurrent mutations.To try to determine whether these variants are technical errors, the authors would ideally sequence plasmid controls using the same protocols that were used to sequence the original samples. Since this is likely infeasible, the authors should also check their data to see if variation is present at M2-77, NP-G384R, and other apparent sites of recurrent mutation at frequencies below the variant-calling threshold. If variation is present at these sites in most samples at a frequency much higher than in neighboring sites, then this variation may reflect technical errors in sequencing. The authors called variants after mapping reads to a reference genome, but they might also try remapping to the sample consensus sequence to reduce the risk that mapping artifacts are causing these recurrent variant calls.

Following the reviewer’s suggestion, we did not find that there was any significant difference in frequencies of iSNVs below the variant-calling threshold between those found in the recurrent sites and neighbouring sites that are ±10 nucleotide positions away (Figure 4 —figure supplement 2). We also remapped the sample reads to their respective consensus sequence and still detected these recurrent calls across multiple individuals and timepoints at similar frequencies when mapped to a reference genome (Figure 4 —figure supplement 3 v. figure supplement 1C-D).

Line 638: “We also checked that the distributions of frequencies for iSNVs found in recurrent mutation sites (i.e. NP-384 and M2-77) that are below variant calling threshold are comparable to those found in their neighbouring sites (±10 nucleotide positions; Figure 4 —figure supplement 2). Furthermore, we remapped the sample reads to their respective consensus sequences to minimize mapping of technical artefacts. We were still able to detect the recurring NP-G384R and M2-R77* amino acid mutations in multiple individuals and timepoints at similar frequencies when mapped to the reference genome (Figure 4 —figure supplement 1C-D and 3). As such, these recurrent mutations are unlikely to have been resulted from erroneous variant calls of artefacts.”

2. The authors write that non-synonymous mutations accumulate later in patient infections, but it's not clear to me how strongly the data in Figure 2A supports that argument, particularly given the limited number of samples collected at any given day following symptom onset. Does nonsynonymous diversity tend to increase over time in successive timepoints collected from the same individual? Does noise in variant frequencies account for the apparent fluctuations in synonymous diversity over time, and if so, how does noise affect the interpretation of nonsynonymous diversity?

To minimize the effect noise, we performed linear regression on the computed evolutionary rates in Figure 2A for each A/H3N2 infected individual with at least three sampled timepoints (n=39). Among these, 25 of them (64%) had a positive correlation in nonsynonymous iSNV accumulation rates against time (Figure 2 —figure supplement 3). For individuals where samples were only collected at two timepoints (n=8, between D4 to D8 post symptom onset), 6 of them had higher accumulation rates of nonsynonymous variants in the successive sample.

In contrast, synonymous evolutionary rates were negatively correlated against time for 27 (69%) of the 39 A/H3N2 infected individuals with at least three sampling timepoints. If noise was the primary determinant for fluctuations in diversity over time, we should observe similar effects on the trends of nonsynonymous and synonymous variants over time. However, the contrasting patient-specific temporal trends between the two type of variants supports our conclusion that nonsynonymous mutations accumulated later in time when within-host virus population is large enough in size such that mutation-selection balance is reached.

Line 175: “To ensure that this temporal trend was not due to aggregated effects across multiple individuals, we performed linear regression on the computed evolutionary rates for each A/H3N2 infected individual with at least three sampling timepoints (n=39). Nonsynonymous evolutionary rates were positively correlated against time for 25/39 individuals (64%; Figure 2 —figure supplement 3). In contrast, synonymous evolutionary rates were negatively correlated against time for 27 (69%) individuals.”

3. In the simulations of A/H1N1pdm09 in Figure 6A and 6B, the authors simulate a distribution of fitness effects in which 1% of nonsynonymous mutations are neutral and the remaining mutations are weakly (s=0.01) or strongly (s=0.1) beneficial (lines 785-6). This distribution of fitness effects seems unrealistic to me – even for an emerging virus that may be adapting to a new host, most nonsynonymous mutations will still be deleterious because they affect basic protein functions. It's not clear to me that the large increase in nonsynonymous diversity that the authors observe from these simulations would be observed if deleterious mutations were adequately accounted for; the distribution would probably look much more similar to Figures 6D and 6E.

We agree that the distribution of fitness effects that was previously used to simulate A/H1N1pdm09 virus dynamics needed to be reworked. However, as in our response to reviewer 1 on the lack of statistical power in the A/H1N1pdm09 virus dataset. Since we cannot reliably characterise the temporal trends of nonsynonymous and synonymous iSNVs in A/H1N1pdm09 infected individuals, we have now removed simulation results for A/H1N1pdm09 viruses.